# Aircraft observations of gravity wave activity and turbulence in the tropical tropopause layer: prevalence, influence on cirrus clouds, and comparison with global storm-resolving models

**Rachel Atlas[1] and Christopher S. Bretherton[1,2]**

[1]Department of Atmospheric Sciences, University of Washington, Seattle, WA, USA
[2]Allen Institute for Artificial Intelligence, Seattle, WA, USA

**Correspondence:** Rachel Atlas (rachel.atlas@lmd.ipsl.fr)

**Abstract.** The tropical tropopause layer (TTL) is a sea of vertical motions. Convectively generated gravity waves create vertical winds on scales of a few to thousands of kilometers as they propagate in a stable atmosphere. Turbulence from gravity wave breaking, radiatively driven convection, and Kelvin–Helmholtz instabilities stirs up the TTL on the kilometer scale. TTL cirrus clouds, which moderate the water vapor concentration in the TTL and stratosphere, form in the cold phases of large-scale ($> 100$ km) wave activity. It has been proposed in several modeling studies that small-scale ($< 100$ km) vertical motions control the ice crystal number concentration and the dehydration efficiency of TTL cirrus clouds. Here, we present the first observational evidence for this.

High-rate vertical winds measured by aircraft are a valuable and underutilized tool for constraining small-scale TTL vertical wind variability, examining its impacts on TTL cirrus clouds, and evaluating atmospheric models. We use 20 Hz data from five National Aeronautics and Space Administration (NASA) campaigns to quantify small-scale vertical wind variability in the TTL and to see how it varies with ice water content, distance from deep convective cores, and height in the TTL.

We find that 1 Hz vertical winds are well represented by a normal distribution, with a standard deviation of $0.2$–$0.4$ m s$^{-1}$. Consistent with a previous observational study that analyzed two out of the five aircraft campaigns that we analyze here, we find that turbulence is enhanced over the tropical west Pacific and within 100 km of convection and is most common in the lower TTL (14–15.5 km), closer to deep convection, and in the upper TTL (15.5–17 km), further from deep convection.

An algorithm to classify turbulence and long-wavelength (5 km $< \lambda <$ 100 km) and short-wavelength ($\lambda <$ 5 km) gravity wave activity during level flight legs is applied to data from the Airborne Tropical TRopopause EXperiment (ATTREX). The most commonly sampled conditions are (1) a quiescent atmosphere with negligible small-scale vertical wind variability, (2) long-wavelength gravity wave activity (LW GWA), and (3) LW GWA with turbulence. Turbulence rarely occurs in the absence of gravity wave activity.

Cirrus clouds with ice crystal number concentrations exceeding $20$ L$^{-1}$ and ice water content exceeding $1$ mg m$^{-3}$ are rare in a quiescent atmosphere but about 20 times more likely when there is gravity wave activity and 50 times more likely when there is also turbulence, confirming the results of the aforementioned modeling studies.

Our observational analysis shows that small-scale gravity waves strongly influence the ice crystal number concentration and ice water content within TTL cirrus clouds. Global storm-resolving models have recently been run with horizontal grid spacing between 1 and 10 km, which is sufficient to resolve some small-scale gravity wave activity. We evaluate simulated vertical wind spectra (10–100 km) from four global storm-resolving

simulations that have horizontal grid spacing of 3–5 km with aircraft observations from ATTREX. We find that all four models have too little resolved vertical wind at horizontal wavelengths between 10 and 100 km and thus too little small-scale gravity wave activity, although the bias is much less pronounced in global SAM than in the other models. We expect that deficient small-scale gravity wave activity significantly limits the realism of simulated ice microphysics in these models and that improved representation requires moving to finer horizontal and vertical grid spacing.

## 1   Introduction

Time mean vertical motions in the tropical tropopause layer (TTL) are less than $1\,\mathrm{cm\,s^{-1}}$ (Ortland and Alexander, 2014), and synoptic-scale vertical motions on scales exceeding 100 km are typically less than $10\,\mathrm{cm\,s^{-1}}$ (Sect. 2.1). However, even well away from deep convective updrafts, gravity waves and turbulence can locally produce vertical winds often exceeding $1\,\mathrm{m\,s^{-1}}$, dwarfing the magnitudes of the synoptic-scale winds.

Vertical motions on all scales influence TTL cirrus clouds, which dehydrate the TTL (Jensen et al., 2013). The dehydrated air is then lofted into the stratosphere (Holton et al., 1995). Decreased water vapor in the stratosphere cools Earth's surface and increases stratospheric ozone (Shindell, 2001). It has been estimated that a 1 ppmv (parts per million by volume) increase in stratospheric water vapor has a radiative forcing of $0.24\,\mathrm{W\,m^{-2}}$ (Solomon et al., 2010). TTL cirrus clouds with higher ice crystal number concentrations dehydrate the TTL and stratosphere, and cool Earth's surface, more effectively (Jensen et al., 2013). Thus, TTL cirrus cloud occurrence and microphysical properties together determine the impact of TTL cirrus clouds on climate.

Recently, studies have used observed temperature fluctuations from aircraft (Kim et al., 2016), satellite (Chang and L'Ecuyer, 2020), and balloon measurements (Bramberger et al., 2022) to show that TTL cirrus cloud occurrence is tightly controlled by predominantly large-scale ($> 100\,\mathrm{km}$) wave activity.

Numerous modeling studies have suggested that small-scale ($< 100\,\mathrm{km}$) vertical motions strongly influence TTL cirrus cloud microphysics by initiating new instances of homogeneous freezing (Dinh et al., 2010; Spichtinger and Krämer, 2013; Schoeberl et al., 2015; Jensen et al., 2016; Dinh et al., 2016). However, no existing observational studies have investigated this.

Large-scale wave activity creates large vertical displacements on long timescales and small-scale vertical motions create small vertical displacements on short timescales. On short timescales, cirrus clouds are less able to adjust to rising supersaturations by growing existing ice crystals and are more likely to experience supersaturations exceeding the homogeneous nucleation threshold, forcing them to nucleate new ice particles.

Much of the air in the TTL is highly supersaturated; in the temperature range investigated here (185–210 K), the threshold for homogeneous nucleation is between 1.5 and 2.3 times the ice saturation (Schneider et al., 2021). If updrafts force atmospheric supersaturation beyond these thresholds, then new ice particles will form through homogeneous nucleation. Subsequent downdrafts following homogeneous nucleation will reduce atmospheric saturation. If the timescales are short enough, then the cirrus cloud may not have quenched the atmosphere down to supersaturation, following homogeneous nucleation, and the ice particles may continue growing in the downdrafts. If the cirrus cloud did have time to quench the atmosphere down to supersaturation, then the downdraft will cause sub-saturation. As most TTL cirrus ice particles are of a similar size, this would lead to a reduction in the particle size but is unlikely to significantly decrease the number of ice crystals. Thus, particles homogeneously nucleated in updrafts can persist through downdrafts and irreversibly increase ice crystal number concentrations in cirrus clouds.

Small-scale motions include small-scale gravity wave activity and turbulence. Most of the aforementioned modeling studies have connected small-scale gravity wave activity, in particular, to homogeneous nucleation in TTL cirrus clouds (Spichtinger and Krämer, 2013; Schoeberl et al., 2015; Jensen et al., 2016; Dinh et al., 2016; Podglajen et al., 2018). Fewer studies have considered the effects of turbulence. Dinh et al. (2010) found that radiatively driven convective turbulence in combination with radiatively driven mesoscale circulations helped maintain a simulated TTL cirrus cloud for several days in the absence of strong gravity wave activity. The cirrus cloud in that study, which was 0.5 km thick, with ice water contents of up to $1\,\mathrm{mg\,m^{-3}}$, achieved relatively small turbulence-driven updrafts up to $2.5\,\mathrm{cm\,s^{-1}}$, which is below the noise floor of aircraft vertical wind measurements. However, TTL cirrus clouds can be several kilometers thick, creating larger radiative destabilization over a deeper layer that could induce stronger turbulent updrafts. No existing studies have examined the relative roles of turbulence and gravity wave activity on TTL cirrus cloud microphysics.

Many studies have used vertical wind measurements from aircraft to characterize small-scale motions over the past 2 decades. Most have focused on the midlatitude troposphere (e.g., Gultepe and Starr, 1995; Koch et al., 2005; Muhlbauer et al., 2014), but a series of National Aeronautics and Space

Administration (NASA) flight campaigns gathered high-rate vertical wind measurements in the TTL that are also well-suited for such analysis.

Using data from two of those campaigns, the Airborne Tropical TRopopause EXperiment (ATTREX) in 2013 and 2014, Podglajen et al. (2017) investigated the frequency and characteristics of turbulence in the TTL and estimated its effect on transport. They found that turbulence more commonly occurs closer to deep convection and is most common in the lower TTL ($< 15.5$ km) within 500 km of convection and in the upper TTL ($> 15.5$ km) further away from deep convection. Podglajen et al. (2017) also found evidence suggesting clear-sky sources of turbulence are dominant in the TTL.

Our study shows that these findings also hold for other NASA field campaigns that sampled the TTL in different geographical areas, years, and seasons (Sect. 3.1 and 3.2). It also makes two major new contributions. The first is to distinguish between turbulence and gravity wave activity in the TTL and examine their separate influences on TTL cirrus cloud microphysics (Sect. 3.3). The second is to compare the spatial power spectrum of TTL vertical wind in global storm-resolving models with aircraft measurements over the tropical west Pacific (Sect. 3.4). Section 4 presents our conclusions.

## 2 Preparing the dataset

### 2.1 Aircraft measurements of vertical wind

We analyze data from aircraft campaigns that simultaneously measured ice water content (IWC) and high-rate (sampled faster than 1 Hz) vertical wind in and above the TTL, which we define, for the purpose of this study, as the atmospheric layer between 14 and 18 km altitude and between 30° N and 30° S. Five NASA campaigns meet these criteria, namely the Airborne Tropical TRopopause EXperiment (ATTREX; Jensen et al., 2017) 2013 and 2014 (we treat these 2 different years of ATTREX as two different campaigns), Pacific Oxidants, Sulfur, Ice, Dehydration, and cONvection (POSIDON), the Tropical Composition, Cloud and Climate Coupling experiment (TC4; Toon et al., 2010), and the Cirrus Regional Study of Tropical Anvils and Cirrus Layers – Florida Area Cirrus Experiment (CRYSTAL-FACE; Jensen et al., 2004). TC4 and CRYSTAL-FACE used multiple aircraft, but only data from the WB-57 meet our criteria. We only include data within the TTL or above the TTL in our analysis.

Figure 1 shows flight tracks from these five campaigns, overlaid on time mean outgoing longwave radiation (OLR) from the entire Clouds and the Earth's Radiant Energy System (CERES) level 3 satellite-based dataset (Doelling et al., 2013; NASA/LARC/SD/ASDC, 2017), which spans nearly 22 years. Smaller OLRs indicate more frequent deep convection. The frequency of deep convection during the flight campaigns may be different than for the entire CERES dataset

due to seasonal and interannual variability. ATTREX 2013 is an outlier in that the majority of its sampling was over the tropical east and central Pacific, which has infrequent deep convection. TC4, CRYSTAL-FACE, and POSIDON sampled primarily close to deep convection near Costa Rica, near Florida, and over the tropical west Pacific, respectively. ATTREX 2014 consisted of two transit flights that sampled far from deep convection over the east and central Pacific and six science flights that sampled close to deep convection over the tropical west Pacific.

All campaigns measured vertical wind at 20 Hz using NASA's Meteorological Measurement System (MMS) instrument (Scott et al., 1990). The MMS estimates the vertical wind as the difference between the vertical speed of air relative to the aircraft and the vertical aircraft speed. These are estimated from pressure sensors and aircraft parameters including pitch, roll, and heading. MMS vertical wind data sometimes exhibit discontinuities when the aircraft switches from one maneuver to another (such as from an ascent to a descent) and must be corrected to minimize artifacts caused by this behavior. Following recommendations from Jonathan Dean-Day and Rei Ueyama from NASA, the aircraft data are separated into flight legs or maneuvers, which are stretches of time when the airplane has a near-constant attack angle, and the vertical wind along each flight leg is demeaned and detrended.

Demeaning and detrending does not introduce significant biases for sufficiently long flight legs because, in the TTL, vertical winds averaged over regions of similar size to a flight leg are typically of the order of a centimeter per second. This estimate is based on analyzing vertical winds in the TTL from ERA5 reanalysis averaged over $1° \times 1°$ or about $100$ km $\times 100$ km boxes (not shown). While ERA5 should probably not be trusted in detail for such a purpose, here it is only used to make an order-of-magnitude estimate. We found that 99.9 % of these boxes have mean vertical wind with magnitudes less than $10$ cm s$^{-1}$, and 62.5 % have mean vertical wind with magnitudes less than $1$ cm s$^{-1}$. The small-scale vertical winds that are the focus of this study have magnitudes of at least $25$ cm s$^{-1}$. Additionally, in the rare event that the mean wind exceeds $10$ cm s$^{-1}$ over a $100$ km segment of the atmosphere, that mean wind is unlikely to be measured well by the aircraft. Flight legs that are less than $100$ km long are removed, as demeaning and detrending those legs could remove some small-scale variability and bias our results.

Figure A1 in Appendix A shows corrected and uncorrected vertical wind data for an example flight, demonstrating both the biases in the uncorrected data and the effectiveness of our data-correction procedure in removing them. Correcting the MMS data is necessary for constraining the magnitude of the vertical wind, but high-frequency ($> 1$ Hz) vertical wind variance is well constrained in the uncorrected data because the biases in vertical wind have frequencies lower than 1 Hz.

Throughout this study, we use the vertical wind variance as a proxy for turbulence (Atlas et al., 2020). Many studies (e.g.,

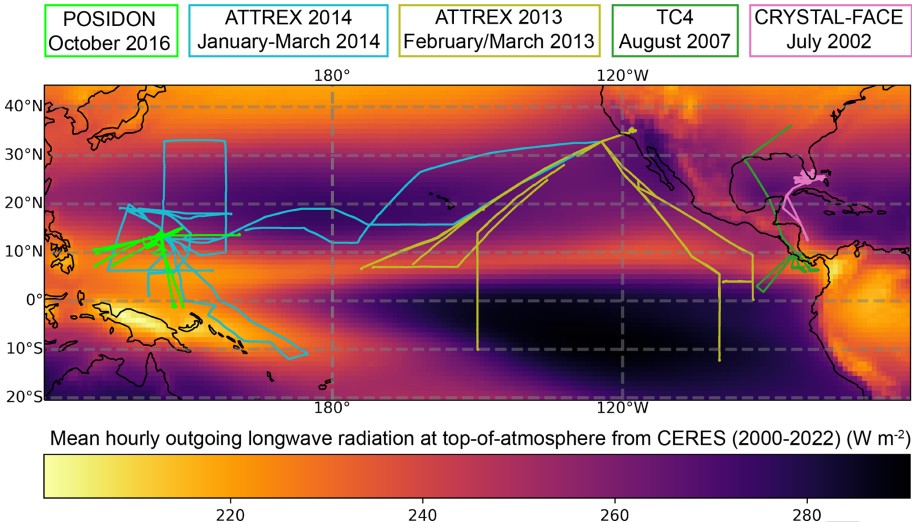

**Figure 1.** Map of mean top-of-atmosphere outgoing longwave radiation from CERES, with flight tracks from the five campaigns used in this study overlaid.

Gultepe and Starr, 1995; Muhlbauer et al., 2014; Podglajen et al., 2017) have used an estimate of the turbulent eddy dissipation rate ($\epsilon$) instead of vertical wind variance to identify and quantify turbulence. $\epsilon$ must be computed from a power spectrum of the aircraft vertical wind. The estimate of $\epsilon$ is sensitive to the algorithm used to make the power spectrum (e.g., Fourier decomposition and wavelet analysis), how $\epsilon$ is estimated from the power spectrum (e.g., fitting a line and integrating the power spectrum), or the assumed Kolmogorov constant. Vertical wind variance is straightforward to calculate and conveys the same information. Within the inertial range of turbulence, it is proportional to $\epsilon^{2/3}$, with a constant of proportionality that is dependent on the aircraft speed and the sampling frequency (Gultepe and Starr, 1995). Figure A1d–e show the vertical wind variance and the reported values of $\epsilon$ for an example flight to show that the same turbulent patches are clearly evident in both metrics.

## 2.2  Aircraft measurements of TTL cirrus cloud microphysics

For all campaigns analyzed, ice water content (IWC) is computed as a difference between total water and water vapor. The cloud laser hygrometer (CLH; Davis et al., 2007) measured total water, and a combination of the Harvard Lyman-$\alpha$ (Weinstock et al., 1994) and the Jet Propulsion Laboratory (JPL) laser hygrometer (May, 1998) measured water vapor during CRYSTAL-FACE and TC4. The reported IWC has a detection limit of about $10^{-1}\,\mathrm{mg\,m}^{-3}$. The NOAA water instrument (Thornberry et al., 2015) measured both total water and water vapor during POSIDON and ATTREX 2013 and 2014. The reported IWC has a detection limit close to $2 \times 10^{-3}\,\mathrm{mg\,m}^{-3}$. Nonzero IWCs that are below the detection limit are uncertain.

IWC and ice crystal number concentration (NI) are tightly linked in TTL cirrus clouds, which tend to have ice size distributions that are more similar than their ice crystal number concentrations. Figure 2a shows percentiles of IWC binned by NI from POSIDON and ATTREX 2014. NI was measured by the Fast Cloud Droplet Probe (FCDP; for particles 3–24 µm; Lance et al., 2010) and the Two-Dimensional Stereo probe (2D-S; for particles 25–3005 µm; Lawson et al., 2006) for both campaigns. The detection limits for the FCDP and the 2D-S, for an aircraft speed of $100\,\mathrm{m\,s}^{-1}$ and assuming sample areas of 0.03 and $80\,\mathrm{mm}^2$, respectively, are about 33 and $0.1\,\mathrm{L}^{-1}\,\mathrm{s}^{-1}$. Median IWC varies nearly linearly with median NI, and the median ice crystal size, assuming single-sized spherical ice particles, varies between about 19 µm (dashed magenta line) and 24 µm (dashed cyan line).

IWC and NI have been quality checked and compiled into a single dataset (Krämer et al., 2020a), as described in Krämer et al. (2020b). Since ATTREX 2013 sampled mainly clear sky, it was not included in the analysis by Krämer et al. (2020b). We still use IWC from this campaign, but we note that it has not been subjected to the same quality control as the other campaigns.

### Distance from convection

Deep convection influences TTL dynamics by generating gravity waves, so it is useful to look at vertical wind variability as a function of the distance to deep convective cores. We estimate the minimum distance to a deep convective core for each 1 Hz sample of aircraft data using the brightness temperature from National Centers for Environmental Prediction (NCEP) Climate Prediction Center's (CPC) merged infrared product (MERGIR; Janowiak et al., 2017). This product has 5 km spatial resolution and 30 min temporal resolution (with

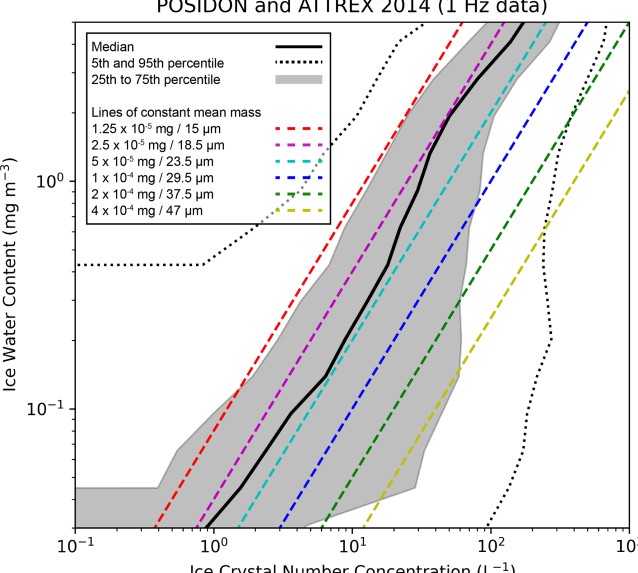

**Figure 2.** Statistics of ice water content (IWC) are shown as a function of ice crystal number concentration (NI) for 1 Hz data from POSIDON and ATTREX 2014. The solid line is the median, the dotted lines are the 5th and 95th percentiles, and the shaded area spans the range of the 25th to 75th percentiles. Colored dashed lines indicate constant mean particle mass. The legend lists the mean masses and the corresponding mean equivalent sphere radii for each colored line.

output on the hour and half-hour). We only use the data on the half-hour because data are frequently missing on the hour. We define convective cores as having brightness temperature below 210 K (Gasparini et al., 2021). For every second of aircraft data, we take the snapshot from MERGIR that is closest in time to the aircraft data, and we compute the distance from the aircraft location to the nearest deep convective core in that snapshot, as illustrated in Fig. A2. Other studies have used different brightness temperature thresholds to identify deep convection. Podglajen et al. (2017) used 235 K and Wall et al. (2020) used a stricter 200 K. We compare these different thresholds in Appendix A and Fig. A2. There we find that a 200 K threshold misses most convective cores entirely, whereas a 235 K threshold includes some anvil cirrus clouds and aging deep convective cores, which are less likely to generate gravity waves.

## 3 Results

### 3.1 Small-scale vertical wind variability in all campaigns

We analyze the 1 Hz vertical wind ($\overline{w}_1$) in this section and high-frequency ($> 1$ Hz) vertical wind variance ($\sigma^2 w_1$) in the next section for all five flight campaigns separately. $\overline{w}_1$ is sensitive to both gravity wave activity and turbulence, whereas $\sigma^2 w_1$ is sensitive mainly to turbulence.

Throughout these two sections, we split the data up into categories based on IWC, distance to deep convection cores, and height. Figure B1 shows distributions of these variables for the five flight campaigns, and Appendix B discusses our choice of categories. We split the IWC into three categories, i.e., clear sky (IWC = 0.0), low-IWC cirrus clouds (between 0.0 and 1 mg m$^{-3}$), and high-IWC cirrus clouds (IWC > 1 mg m$^{-3}$). Clear sky and low-IWC cirrus clouds cannot be perfectly discriminated by the measurements, particularly for CRYSTAL-FACE and TC4, which have an IWC detection limit of 0.1 mg m$^{-3}$.

Figure 3 shows probability distributions of $\overline{w}_1$ for each campaign separately. The first column shows the distribution for all campaign data with the campaign name and the number of 1 Hz samples included in the analysis printed over the plots. The second, third, and fourth columns show probability distributions of $\overline{w}_1$ split into categories based on the IWC, distance to deep convective cores, and height in the TTL, respectively, with pie charts showing the distribution of data across the different categories. Normal distributions are fitted to all probability distributions and the fitted standard deviations ($\sigma$) are printed on the plots.

Values of $\sigma$ vary between 0.17 and 0.4 m s$^{-1}$ for all campaign data (first column of Fig. 3). Differences in vertical wind variability across the set of campaigns may arise from sampling closer to or farther from deep convection, the properties of the deep convection (e.g., land vs. ocean and shallow vs. deep), interannual, seasonal, and geographic variability in the upper troposphere and TTL, including the quasi-biennial oscillation (QBO) phase, and sampling different heights within the TTL. CRYSTAL-FACE, which sampled near-convection in the Florida region, has the widest distribution of $\overline{w}_1$ (the most vertical wind variability). ATTREX 2013, which sampled the tropical east and central Pacific, usually far from deep convection, has the least variability. Because the five campaigns sampled such different conditions, it is plausible that they approximately span the vertical wind variability to be expected anywhere in the TTL outside of the immediate vicinity of deep convection.

The remaining columns of Fig. 3 partition the data from each experiment into categories of IWC, distance from deep convection, and height and compare the distributions of $\overline{w}_1$ across these categories. Within ATTREX 2014, POSIDON, and CRYSTAL-FACE data, high-IWC cirrus clouds have wider $\overline{w}_1$ distributions (top three rows of second column of Fig. 3), and this result is insensitive to the threshold used to define high-IWC cirrus clouds (not shown). In general, vertical wind variability increases closer to deep convection (third column of Fig. 3). An exception is ATTREX 2013, where vertical wind variability is low, regardless of the distance from deep convection. The first column of Fig. B2 shows the same analysis for clear-sky data only; the results are the same. Thus, the increase in vertical wind variability close to deep convection is not caused by a higher occurrence of cirrus cloud close to deep convection. There is no consistent re-

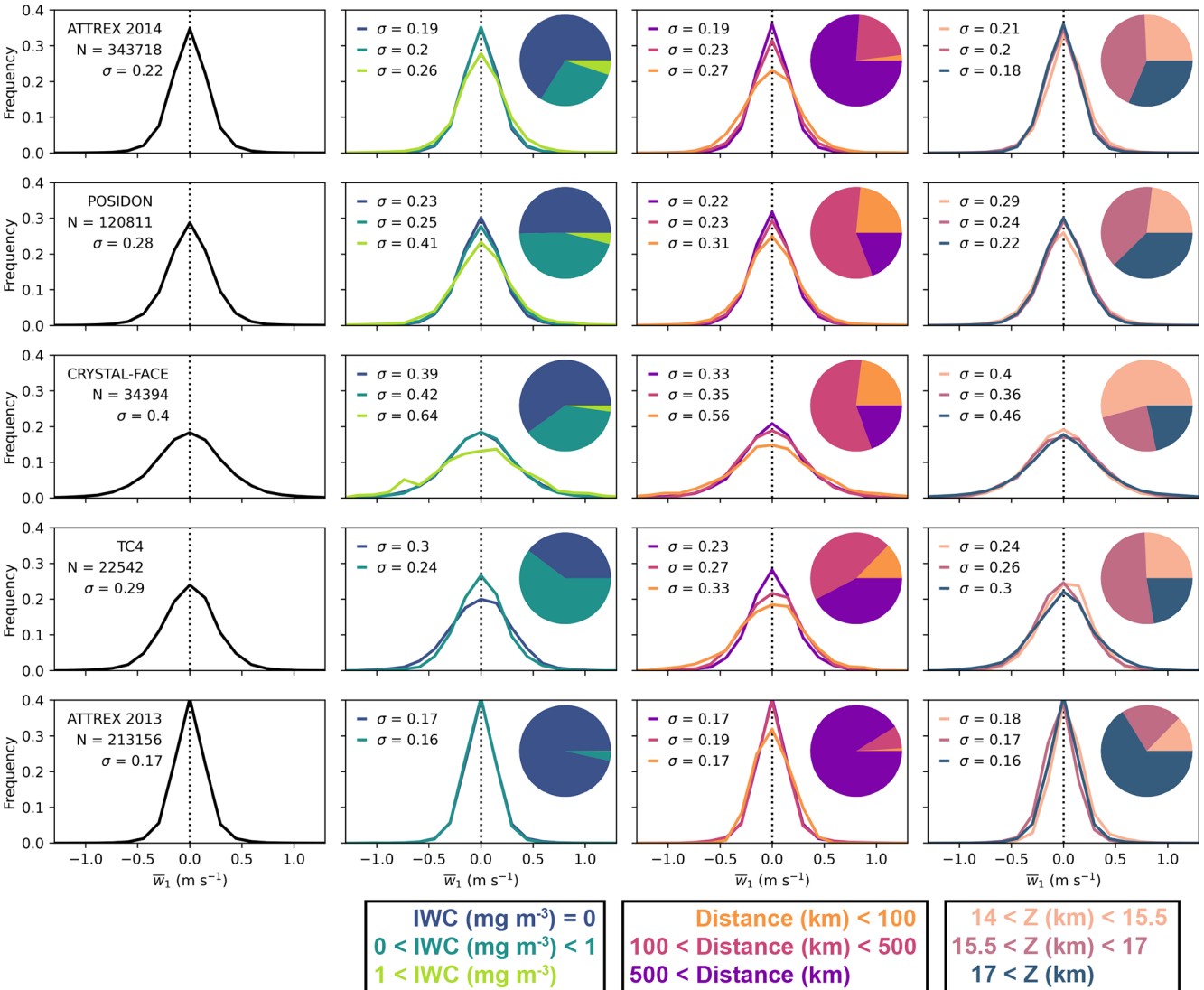

**Figure 3.** Distributions of 1 Hz mean vertical wind ($\overline{w}_1$), for corrected data from all campaigns, shown in separate rows. Campaign names and the number of 1 Hz samples are given in the leftmost column. Second, third, and fourth columns split the data into categories (relative frequencies are shown in pie charts) based on IWC, distance from deep convection, and height in the TTL, respectively. Standard deviations of fitted normal distributions ($\sigma$) are printed on all panels.

lationship between vertical wind variability and height across the five campaigns or between the analysis of all data (Fig. 3) and clear-sky data (Fig. B2).

## 3.2  Turbulence in all campaigns

Figure 4 shows distributions of high-frequency ($> 1$ Hz) vertical wind variance ($\sigma^2 w_1$). $\sigma^2 w_1$ is the variance of the 20 sub-samples within each second of data and a proxy for turbulence. The distributions are split into the same categories based on IWC, distance from deep convection, and height, as in Fig. 3. We define $\sigma^2 w_1$ greater than an empirical detectability threshold of $0.01\,\mathrm{m^2\,s^{-2}}$ as turbulent. This turbulence threshold is typically lower than the one

used by Podglajen et al. (2017), who analyzed strong turbulence rather than just detectable turbulence. This can be seen by comparing the shaded yellow regions of the flight tracks shown in Fig. A1d and e. Our methodology therefore flags turbulence with a much higher frequency. To facilitate comparison with Podglajen et al. (2017) and test the robustness of our results to the turbulence threshold used, Fig. B3 shows a version of Fig. 4 but with distributions of $\epsilon$ instead of $\sigma^2 w_1$ and with a turbulence threshold for $\epsilon$ of $10^{-3}\,\mathrm{m^2\,s^{-3}}$. The sensitivity of turbulence to environmental categories is qualitatively similar to Fig. 4, but the implied turbulence frequency is an order of magnitude smaller. We would need to increase our threshold high-frequency vertical wind variance to over $0.01\,\mathrm{m^2\,s^{-2}}$ (not shown) to obtain comparable results.

We choose our comparably weak detectability threshold because we aim to associate all vertical wind anomalies exceeding $25 \, \mathrm{cm \, s^{-1}}$ with a particular type of small-scale vertical wind variability. Using a strong turbulence threshold would result in many of these periods going undetected. Additionally, the infrequency of strong turbulence would result in poorer sampling statistics.

Using our variance-based algorithm, the frequency of turbulence varies between about 6 % and 12 % for all campaign data (first column of Fig. 4). ATTREX 2014 and POSIDON, which both sampled the TTL mainly above the tropical west Pacific, have about twice as frequent turbulence as CRYSTAL-FACE and TC4, even though the latter campaigns have more vertical wind variability. Thus, turbulence accounts for a larger proportion of the vertical wind variability sampled during ATTREX 2014 and POSIDON than during TC4 and CRYSTAL-FACE.

The frequency of turbulence for clear-sky data (dark blue line; second column in Fig. 4) varies between about 5 % and 10 % and is typically just slightly lower than the all-sky frequency, implying that clear-air turbulence accounts for much of the turbulence in the TTL. The frequency of turbulence for high-IWC cirrus clouds is at least 3 times that for clear-sky data (second column in Fig. 4). High-IWC cirrus clouds have more turbulence than clear-sky data, independently of the threshold used for turbulence and the threshold used to define high-IWC cirrus clouds (not shown).

Increased turbulence can account for much of the widening of the vertical wind distribution within high-IWC cirrus clouds (seen in Fig. 3) for ATTREX 2014, POSIDON, and CRYSTAL-FACE.

In all campaigns, turbulence is least frequent further than $500 \, \mathrm{km}$ away from convection (third column of Fig. 4), which is consistent with the findings of Podglajen et al. (2017). In all campaigns, except TC4, turbulence is most frequent within $100 \, \mathrm{km}$ of convection. Column 3 in Fig. B2 shows that the same is true for clear-sky data. However, the differences in the frequency of turbulence between the two other categories ($100$–$500 \, \mathrm{km}$ and $> 500 \, \mathrm{km}$) are largely diminished in the clear-sky data. If a stricter threshold is used for turbulence, then the results are the same except for ATTREX 2013, where the differences in the amount of turbulence within $100 \, \mathrm{km}$ of convection, and between $100$ and $500 \, \mathrm{km}$, are diminished (not shown).

Turbulence is enhanced below $15.5 \, \mathrm{km}$ in all campaigns, except ATTREX 2013, which has enhanced turbulence between $15.5$ and $17 \, \mathrm{km}$. Podglajen et al. (2017) found that turbulence was enhanced below $15.5 \, \mathrm{km}$, mainly within $500 \, \mathrm{km}$ of deep convection (Fig. 9 in their paper). Consistent with these results, ATTREX 2013 has the most sampling further than $500 \, \mathrm{km}$ away from deep convection (Figs. 3 and B1). These results are insensitive to the turbulence threshold used.

In this section and in the preceding section, we analyzed aircraft data from the TTL from five NASA aircraft campaigns that took place in different years, seasons, and geo-graphical areas and sampled a different range of distances from deep convection and heights within and above the TTL. Across all campaigns, the probability distribution of $\overline{w}_1$ is well approximated as a normal distribution, with a standard deviation between $0.17$ and $0.56 \, \mathrm{m \, s^{-1}}$, depending on the distance from deep convection, the height in the TTL, and the presence of cloud. Vertical wind variability, which is influenced by both turbulence and gravity wave activity, is largest during CRYSTAL-FACE, but the frequency of turbulence is largest during ATTREX 2014 and POSIDON. That means that the increased vertical wind variability during CRYSTAL-FACE is due to increased gravity wave activity. It is unlikely that these differences are purely due to different sampling strategies across the campaigns, and we encourage future studies to investigate the causes of geographical differences in the frequency of turbulence and small-scale gravity wave activity.

We verified that several findings reported in Podglajen et al. (2017) about the frequency of turbulence in ATTREX 2013 and 2014 are true across our entire set of campaigns, including that turbulence is enhanced closer to deep convection, below $15.5 \, \mathrm{km}$ in the TTL when close to deep convection, and above $15.5 \, \mathrm{km}$ when far away from deep convection. Furthermore, we analyzed turbulence in clear-sky and cloudy data separately and found that turbulence is strongly enhanced within high-IWC TTL cirrus clouds.

## 3.3 Sources of small-scale vertical wind variability during ATTREX

Level (constant-altitude) flight legs are useful for separately detecting turbulence and gravity wave activity and for performing spectral analyses because they do not conflate horizontal and vertical scales of variability. Gravity waves often have smaller vertical wavelengths than horizontal wavelengths (Bramberger et al., 2022), so the scale separation between gravity wave activity and turbulence is more pronounced in the horizontal.

ATTREX 2014's flight strategy involved repeatedly flying parallel to the ground at a height of about $14.2 \, \mathrm{km}$, typically through cloud, and performing a slow ascent and a quick descent (Fig. A1). ATTREX 2013 sometimes employed this flight strategy and sometimes performed ascents and descents with no level legs in between. There are 52 level legs that are at least $100 \, \mathrm{km}$ long in the ATTREX 2014 dataset and 13 in the ATTREX 2013 dataset, which are the focus of the rest of this study. In this section, we present an algorithm to distinguish between turbulence and gravity wave activity, and we investigate how the presence of turbulence and gravity wave activity varies with distance to deep convection, IWC, and NI. In the following section, we perform spectral analyses of vertical wind on level leg data from ATTREX 2014 and simulated vertical winds from four global storm-resolving models from the DYnamics of the Atmospheric general circulation Modeled On Non-hydrostatic Domains

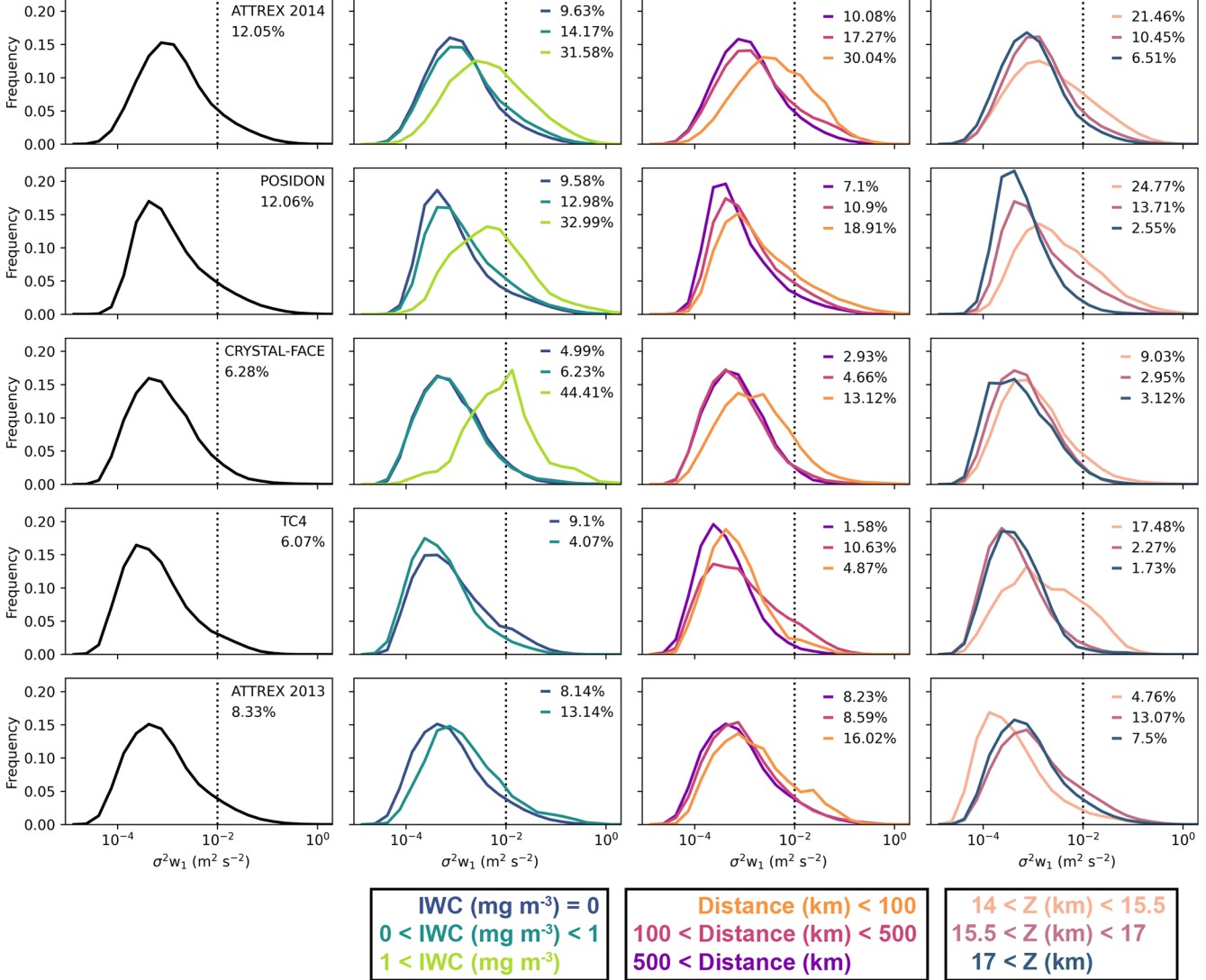

**Figure 4.** Same as Fig. 3 except showing distributions of high-frequency ($> 1$ Hz) vertical wind variance ($\sigma^2 w_1$). The percentage of data exceeding the turbulent threshold ($0.01$ m$^2$ s$^{-2}$; vertical dotted line) for each category is shown on each plot.

summer experiment (DYAMOND-1; Stevens et al., 2019) to evaluate simulated small-scale vertical wind variability over the tropical west Pacific.

Gravity wave activity occurs on a wide range of horizontal scales from 1 km to thousands of kilometers. We define long-wavelength gravity wave activity (LW GWA) as having dominant wavelengths between 5 and 100 km and short-wavelength gravity wave activity (SW GWA) as having smaller wavelengths. The purpose of making this distinction is that SW GWA has horizontal length scales that overlap with turbulence, whereas LW GWA does not.

We separate the level legs from ATTREX into non-overlapping 25 s or 5 km segments (the aircraft travels at about 200 m s$^{-1}$ during level legs). LW GWA occurs on spatial scales larger than 5 km, so we classify each entire level leg into one of two categories, i.e., (1) negligible LW GWA

and (2) LW GWA. All 5 km segments within a level leg receive the same classification.

Turbulence and SW GWA occur on scales smaller than 5 km, so we classify each 5 km segment into one of three different categories, i.e., (1) negligible sub-5 km variability, (2) turbulence, and (3) SW GWA. Because SW GWA and turbulence occur on similar spatial scales, we can only detect turbulence in the absence of SW GWA. We detail the classification of level legs and 5 km segments in Appendix C.

Figure 5 shows four example time series of vertical wind for four different level legs from the ATTREX 2014 dataset. In the upper time series plot for each example, the 20 Hz vertical wind is shown in gray, and the mean vertical wind for the 5 km segments ($\overline{w}_{25}$) is shown in black. In the second row under each example, the 20 Hz vertical wind is color coded according to the sub 5 km variability classifications,

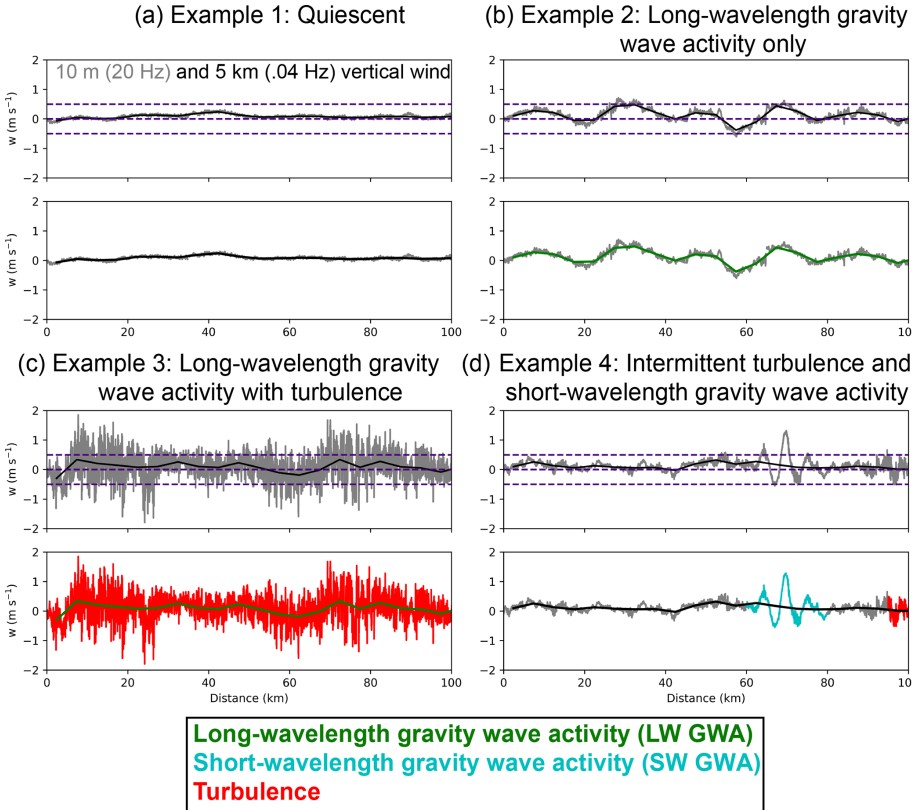

**Figure 5.** Four example vertical wind time series for 100 km long segments from ATTREX 2014 level legs. All plots show the time series of high-rate (20 Hz/10 m) vertical wind and 5 km mean vertical wind ($\overline{w}_{25}$). In the top time series for each example, high-rate vertical wind is in gray and $\overline{w}_{25}$ is in black. Horizontal dashed purple lines indicate 0.0, 0.5, and $-0.5\,\mathrm{m\,s}^{-1}$. In the bottom time series for each example, $\overline{w}_{25}$ is colored green if there is long-wavelength gravity wave activity, and the high-rate vertical wind is colored cyan and red where there is short-wavelength gravity wave activity and turbulence, respectively.

and $\overline{w}_{25}$ is color coded according to the LW GWA classification (for the entire level leg). Examples 2 and 3 have LW GWA, whereas examples 1 and 4 have negligible LW GWA. All of the 5 km segments in examples 1 and 2 have negligible sub 5 km variability, whereas in example 3 they are all turbulent. Example 4 has one intermittent patch of turbulence and one patch of SW GWA. These small-scale vertical motions are typically associated with temperature anomalies of up to 0.5 K for SW GWA and 0.1–0.2 K for turbulence (not shown; for an example, see Fig. 5 in Podglajen et al., 2017).

Each 5 km segment can have three possible sub 5 km classifications, with or without LW GWA. We show the frequency of each combination of classifications in Fig. 6a. Hatching indicates LW GWA, and the color represents the sub 5 km variability classification. The following three situations are most common: (1) negligible LW GWA with negligible sub 5 km variability (quiescent), (2) LW GWA with negligible sub 5 km variability (LW GWA only), and (3) LW GWA with turbulence. These situations occur 24 %, 50 %, and 22 % of the time, respectively. LW GWA is present 75 % of the time. Turbulence occurs preferentially with LW GWA, such that only 5 % of turbulent segments (11 % for clear-sky

data) do not have LW GWA (Fig. 6a). SW GWA also occurs preferentially with LW GWA, as only 8 % of segments with SW GWA do not also have LW GWA (these segments are represented by the thin sliver in cyan between the areas of the pie chart labeled as quiescent and as LW GWA with turbulence).

Figure 6b shows histograms of 5 km mean (from top to bottom) NI, IWC, and distance to deep convection. The extra column on the left-hand side of the histograms for NI and IWC corresponds to NI = 0 and IWC = 0, respectively, and the black lines show thresholds defining high NI ($> 20\,\mathrm{L}^{-1}$) and high IWC (CE1) ($> 1\,\mathrm{mg\,m}^{-3}$), which is also used in the category definitions of Fig. 6c. Segments classified as LW GWA with turbulence are most common, and segments classified as quiescent are rarely seen within high-NI and high-IWC cirrus clouds or within 500 km of convection (to the left of the black line on the histogram for the distance to deep convection).

Figure 6c shows pie charts of 5 km mean NI, IWC, and distance from deep convection for the three most common conditions from Fig. 6a, with categories used to simplify the analysis. There are fewer total samples in the pie charts for

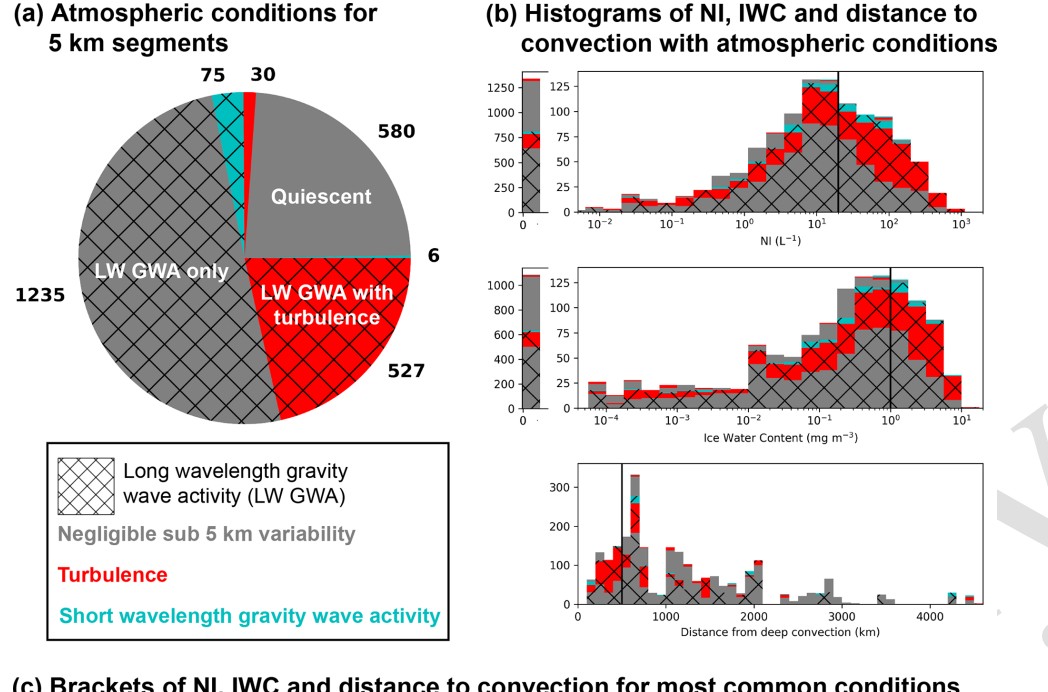

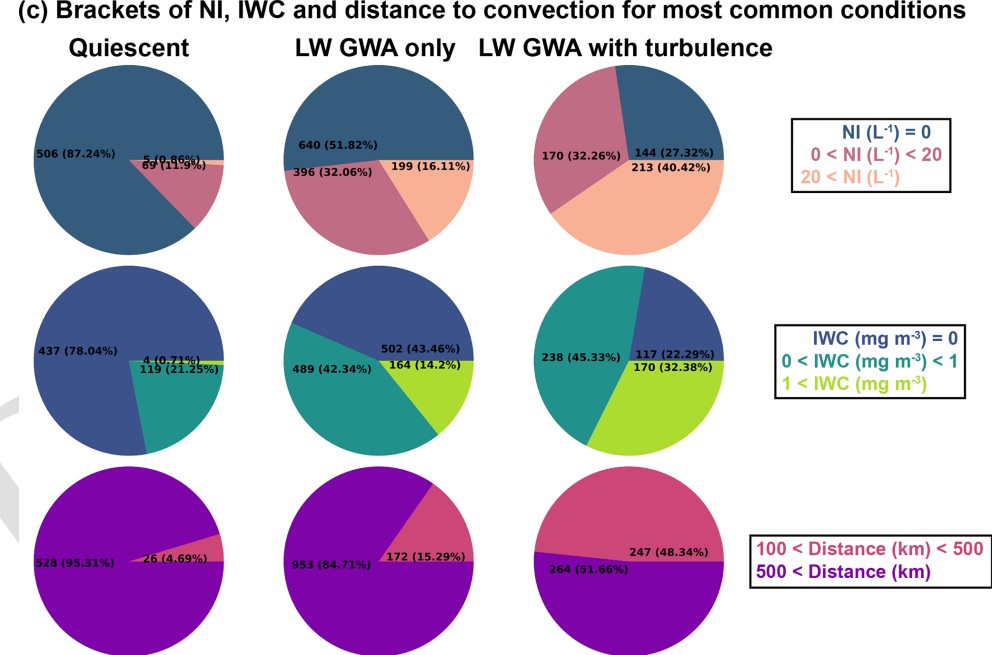

**Figure 6.** Analysis of 5 km segments from ATTREX 2013 and 2014 level legs. **(a)** Frequency of atmospheric conditions. The most common conditions are labeled. **(b)** Histograms of (top to bottom) NI, IWC, and distance from deep convection, showing the contributions from different atmospheric conditions. Black lines indicate thresholds also used below in panel **(c)**. **(c)** Pie charts showing the relative frequency of categories of (top to bottom) NI, IWC, and distance to deep convection for the three most frequent atmospheric conditions.

the distance to deep convection than there are in the pie charts for IWC and NI because brightness temperature data are occasionally missing over the tropical west Pacific.

The likelihoods of occurrence of high-NI cirrus clouds, given quiescent conditions, LW GWA only, and LW GWA with turbulence, are 0.9 %, 16 %, and 40 %, respectively.

Thus, high-NI cirrus clouds are about 20 times more likely when there is gravity wave activity and 50 times more likely when there is also turbulence, compared to quiescent conditions. The likelihoods of occurrence of high-IWC cirrus clouds, given quiescent conditions, LW GWA only, and LW GWA with turbulence, are 0.7 %, 14 %, and 33 %, respec-

tively. Thus, high-IWC cirrus clouds are also about 20 times more likely when there is gravity wave activity and about 50 times more likely when there is also turbulence, compared to quiescent conditions. High-NI and high-IWC cirrus clouds co-occur with low-frequency gravity wave activity 99 % of the time.

The fractions of 5 km segments within 500 km of convection with quiescent conditions, LW GWA only, and LW GWA with turbulence are 5 %, 15 %, and 47 %, respectively. Thus, turbulence is enhanced closer to deep convection to a much greater extent than LW GWA.

Turbulence co-occurs with high-NI cirrus clouds 41 % of the time and with high-IWC cirrus clouds 32 % of the time (Fig. 6c). However, turbulence is not a necessary condition for high-NI and high-IWC cirrus clouds, as they occur in the absence of turbulence about half of the time. In general, the presence of turbulence is much more highly correlated with the presence of LW GWA than with the presence of high-NI and high-IWC cirrus clouds.

## 3.4 Evaluating vertical wind variability in global storm-resolving models

Recently, advances in computing power have made it possible to run global atmospheric models with horizontal grid spacing below 5 km. These models are referred to as global storm-resolving models (GSRMs) because they explicitly resolve deep convection rather than using a deep convective parameterization. Since deep convection is a major source of both gravity waves and the water vapor and ice that form TTL cirrus clouds, this makes GSRMs attractive for studying TTL cloud formation processes, including lifting within gravity waves. However, GSRMs do not resolve turbulence, which they typically parameterize in some form. In this section, we use ATTREX observations to address whether TTL vertical winds simulated by GSRMs are sufficiently realistic to form a credible dynamical environment for TTL cirrus cloud formation and evolution.

Stephan et al. (2019) found that explicit convection simulated with a horizontal grid spacing of 5 km produces more gravity wave momentum flux at 30 km in the tropics and subtropics, where convection is the predominant source of gravity wave activity (Fritts and Alexander, 2003, and references therein), and a wider vertical wind distribution throughout the troposphere, compared to parameterized convection. Müller et al. (2018) found that "convective parameterization inhibits gravity wave generation by convective clouds". While these studies suggest gravity wave generation and propagation are more realistic in GSRMs than in coarse-resolution models, substantial discrepancies may still exist between GSRMs and the real atmosphere.

In the DYAMOND-1 intercomparison, nine GSRMs were identically initialized from reanalysis and run freely (without nudging) for the 40 d period of 1 August–10 September 2016. Here, we focus on four of those models, namely

Nonhydrostatic ICosahedral Atmospheric Model (NICAM), Global System for Atmospheric Modeling (gSAM), Finite-Volume Cubed-Sphere Dynamical Core (FV3), and ICON (Icosahedral Nonhydrostatic Weather and Climate Model). NICAM, gSAM, FV3, and ICON have horizontal grid spacings of 3.25, 5, 3.25, and 2.5 km, respectively. The vertical grid spacing in the TTL is 400 m for NICAM and close to 500 m for the other three models.

Figure 7 shows snapshots of vertical wind from the 141st hour of simulation at the vertical levels closest to 14.2 km, which is the same level as the ATTREX horizontal aircraft legs. Figure S1 in the Supplement is an animation of vertical wind snapshots for hours 48–957 of the simulations (allowing 2 d for model spinup), showing that the snapshot in Fig. 7 is representative of the simulations. The models differ substantially in their magnitudes of vertical winds and the dominant scales of vertical wind variability. gSAM is an outlier in having more vertical wind variability than the other models, particularly at the grid scale. ICON has more grid-scale variability than FV3 and NICAM. NICAM has larger dominant scales of horizontal variability than the other models, and FV3 has weak vertical winds in most grid cells but strong gravity wave activity in the vicinity of deep convection.

Figure 8b shows statistics of power spectra of 20 Hz vertical wind from ATTREX 2014 level legs sampling the tropical west Pacific and from the four GSRMs for the same regions and vertical levels shown in Fig. 7. We perform 1-D fast Fourier transforms (FFTs) for both simulations and observations. FFTs are performed separately along the east–west and north–south direction in the GSRMs and then averaged together. The spectra are similar for both directions (not shown). ICON has an unstructured grid. To enable 1-D FFT analysis, it is interpolated to an 0.01° lat–long grid. For the observations only, we use Welch's method for windowed Fourier analysis with a window length of 100 km and a Hann taper to reduce noise in the spectra.

All simulations have too little power in the vertical wind at all wavelengths examined. The power spectra for ICON, FV3, and NICAM decrease strongly towards the grid scale, which is where they are furthest from the observed spectra. gSAM has a flatter spectrum which more closely resembles the observations. However, the grid-scale variability in gSAM has more resemblance to white noise than to gravity wave activity, so it may not be physically meaningful for this analysis.

We do not look at relationships between cirrus cloud properties and vertical wind here because none of the models predict NI, and while all models predict IWC, FV3 outputs it on a different grid than the vertical wind, making comparisons difficult. Nonetheless, our observational analysis showed that small-scale gravity wave activity is a strong control on TTL cirrus cloud microphysics, and our comparison here shows that small-scale gravity wave activity is underrepresented in the evaluated GSRMs. If the GSRMs have enough or too

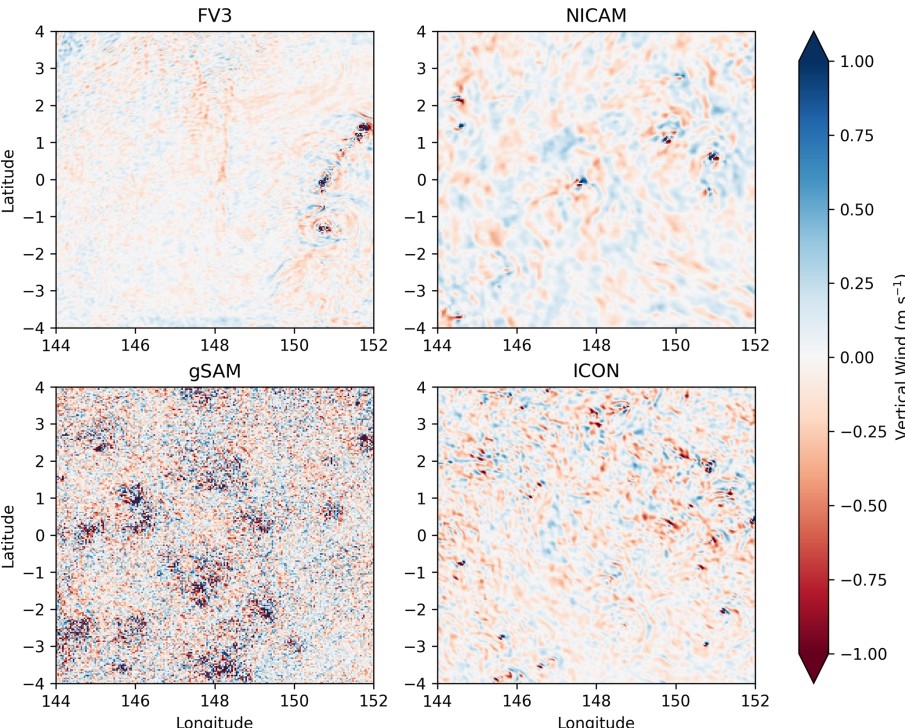

**Figure 7.** Snapshots of vertical wind from hour 141 in the DYAMOND-1 simulations. Winds are taken from the model level closest to 14.2 km, and the tropical west Pacific region is shown.

much ice within simulated TTL cirrus clouds compared to the real atmosphere, then that suggests that overproduction of ice within the microphysics schemes is compensating for deficient dynamical contributions to ice production. In this case, the physical mechanisms controlling ice production in simulated TTL cirrus clouds would differ substantially from the real atmosphere.

## 4   Conclusions

Tropical tropopause layer (TTL) cirrus clouds can be influenced by small-scale vertical motions in the TTL from gravity wave activity and turbulence. The relationships between these phenomena are analyzed using high-rate vertical wind data collected by NASA flight campaigns.

Out of the five campaigns we analyzed, vertical wind variability was largest during CRYSTAL-FACE and TC4, although those campaigns had the lowest frequencies of turbulence, indicating that gravity wave activity was an important source of variability.

Turbulence during ATTREX 2013 and 2014 was analyzed in detail in Podglajen et al. (2017), and we find that some key results from that study hold true across the five campaigns we analyzed; i.e., (1) turbulence is enhanced over the tropical west Pacific and nearer to deep convection, and (2) turbu-

lence is most frequent in the lower TTL (14–15.5 km), close to deep convection, and in the upper TTL (15.5–17 km), further from deep convection.

For the first time, we used aircraft measurements to correlate gravity wave activity and turbulence with TTL cirrus cloud microphysical properties. During ATTREX 2014, 99 % of 5 km segments with high ice water content (IWC > 1 mg m$^{-3}$) and high ice crystal number concentrations (NI > 20 L$^{-1}$) co-occurred with long-wavelength gravity wave activity, and half of those segments co-occurred with turbulence as well. Thus, small-scale vertical motions driven by turbulence and gravity wave activity are key to producing thicker cirrus clouds that contain more ice crystals over the tropical west Pacific.

A strong relationship between small-scale vertical wind variability and TTL cirrus cloud microphysics had been proposed in several modeling studies, but here we present the first observational evidence for it. Each of the modeling studies focused on either gravity wave activity or turbulence. Here, we show that both sources of small-scale vertical wind variability are important and that they frequently occur together.

The common co-occurrence of thicker cirrus clouds and turbulence can be explained in the following two ways: (1) thicker clouds initiate cloud-driven turbulence, and

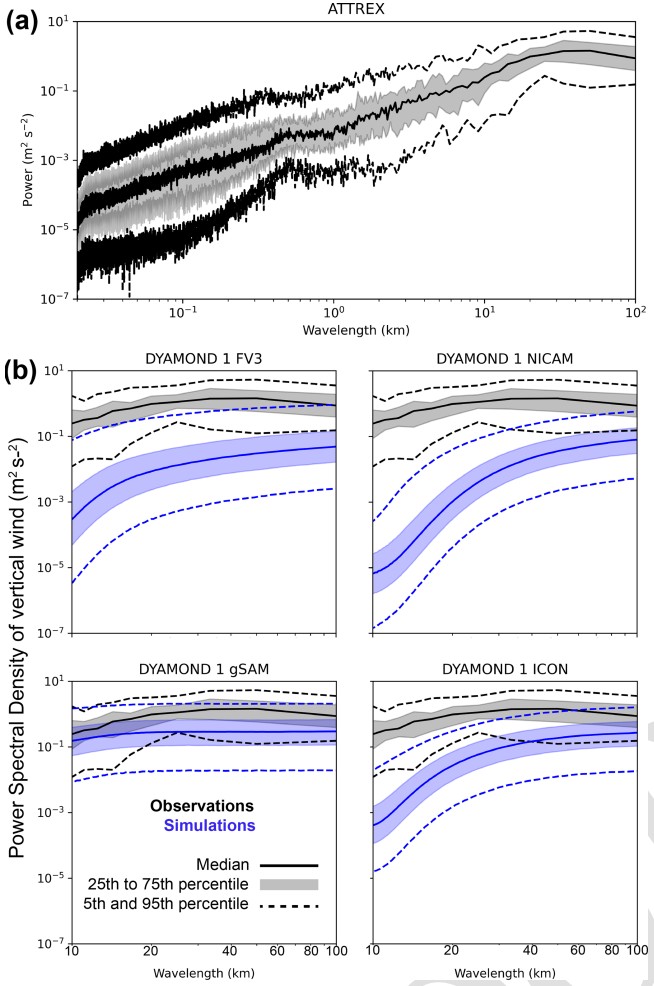

**Figure 8. (a)** Statistics of 1-D power spectral density for the AT-TREX 2014 observations over the tropical west Pacific for all measured wavelengths. The increase in power at ∼ 500 m is caused by the oscillation of the aircraft. **(b)** Statistics of 1-D power spectral density for the ATTREX 2014 observations over the tropical west Pacific (black) and the DYAMOND-1 simulations (blue) for wavelengths between 10 and 100 km. Solid lines are medians, dotted lines are 5th and 95th percentiles, and shaded areas span the ranges of the 25th to 75th percentiles.

(2) clear-air turbulence forms cirrus clouds and/or thickens pre-existing cirrus clouds. These explanations are not mutually exclusive. Our analysis cannot rule out the possibility that cloud-driven turbulence occurs in the TTL, but there are several clues from our study and from Podglajen et al. (2017) indicating that clear-sky sources of turbulence are dominant.

Podglajen et al. (2017) found that turbulent patches were correlated with a low Richardson number, indicating the presence of shear. Shear can cause two types of clear-air turbulence, namely gravity wave breaking, by creating a critical level, and Kelvin–Helmholtz instabilities. Additionally, gravity wave breaking can create or strengthen shear layers (Dörnbrack, 1998). Here, we found that clear-air turbulence

was common in all five flight campaigns analyzed and was enhanced closer to deep convection, which also is a source of gravity wave activity and layers of locally enhanced vertical wind shear. During ATTREX, turbulence co-occurred with gravity wave activity 95 % of the time and thicker cirrus clouds only 30 %–40 % of the time. We encourage future studies to more closely examine turbulent sources in the TTL.

Another potential source of vertical wind variability that we were not able to examine within this study is cloud-driven mesoscale circulations. Cloud-driven mesoscale circulations and cloud-driven turbulence are both induced by thermal instabilities in cirrus clouds, but they produce vertical motions on different scales. Several modeling studies have suggested that cloud-driven mesoscale circulations can maintain cirrus clouds (Dobbie and Jonas, 2001; Dinh et al., 2010; Jensen et al., 2011), whereas at least one other has found that they cannot (Boehm et al., 1999). However, none of these modeling studies included gravity wave activity. We encourage future modeling studies to analyze the development and influence of cloud-driven mesoscale circulations and turbulence in the presence of realistic gravity wave activity.

We also compared vertical wind variability simulated by global storm-resolving models (GSRMs) in the lower TTL over the tropical west Pacific with ATTREX 2014 data. The four models we evaluated (gSAM, ICON, FV3, and NICAM) had drastically different magnitudes of vertical wind and scales of vertical wind variability. Only gSAM had variability at wavelengths shorter than 100 km comparable to the observations. Thus, GSRMs underestimate the vertical winds that affect TTL cirrus clouds, with potential impacts on their simulated microphysics.

Many aspects of the model dynamics and the experimental setup may affect gravity wave formation and propagation, but the horizontal and vertical resolution are likely limiting factors. The effective resolution (the minimum length scale that can be resolved) may be 6 times the horizontal grid spacing for GSRMs (Caldwell et al., 2021), meaning that only gravity waves with wavelengths larger than 19.5 to 25 km can be supported in the GSRMs in this study. Additionally, studies have found that a vertical grid spacing of 200 m or finer in the upper troposphere is necessary to adequately handle gravity wave propagation and achieve convergence (Kuang and Bretherton, 2004; Skamarock et al., 2019), but the GSRMs in this study have a vertical resolution at least twice as coarse as that in the TTL. Thus, we encourage future GSRMs or, more practically, regional cloud-resolving model studies to examine the effects of increased vertical and horizontal resolution on small-scale vertical wind variability.

# Appendix A: Preparing the dataset: additional information

## A1   Correcting vertical wind data

Figure A1 shows flight tracks for an example flight from AT-TREX 2014. Figure A1b color codes the flight track with the uncorrected vertical wind. After 55 000 s into the flight, a pattern emerges of apparent downdrafts throughout each descent, apparent updrafts in the level leg and the first half of the following ascent, and apparent downdrafts in the second half. This nonphysical behavior indicates that changing aircraft maneuvers (going from an ascent to a descent, for example) are affecting the measured vertical winds. To mitigate this artifact, we demean and detrend each flight leg (each solid color segment in Fig. A1a), and we remove legs that cover a horizontal distance smaller than 100 km, to produce the corrected data shown in Fig. A1c. The magnitudes of the corrected vertical winds are smaller and less skewed toward negative values.

Figure A1d–e show the high-frequency ($> 1$ Hz) vertical wind variance ($\sigma^2 w_1$) and the 1 s turbulent kinetic energy dissipation rate ($\epsilon$), as reported in the NASA dataset, respectively. $\epsilon$ has rare outliers with unrealistically high values outside patches of turbulence, which are mainly seen between 60 000 and 65 000 s into the flight (red points). Podglajen et al. (2017) did not use the reported $\epsilon$, so their analysis was not affected by these outliers.

Outliers aside, the same turbulent patches are evident in both $\epsilon$ and $\sigma^2 w_1$. Both metrics are similarly useful for identifying turbulence; we chose $\sigma^2 w_1$ based on its ease of calculation and interpretation. In Fig. A1d and e, the yellow color indicates data that are identified as turbulent in this study and in Podglajen et al. (2017), respectively. Our turbulence threshold is lower than the one used in Podglajen et al. (2017), so we identify a larger percentage of the data as being turbulent.

Figure A1 also demonstrates the sampling strategy for AT-TREX 2014 and parts of ATTREX 2013, which included many level legs that are the focus of the analyses in Sect. 3.3 and 3.4.

## A2   Computing distance from convection

Figure A2 demonstrates how we compute the distance to deep convection for the aircraft data and compares different brightness temperature thresholds for identifying deep convection. We are interested in deep convective cores as sources of gravity wave activity, moisture, and possible detrained ice in the TTL. Only one pixel in all four snapshots has a brightness temperature below 200 K (marked with a green star in the upper right plot), so that threshold is too strict. A threshold of 235 K (white contours) includes some outflow cirrus clouds and remnants of deep convection, which are areas that are influenced by deep convection but that are less likely to generate gravity waves. We select an intermediate value of 210 K (pink points in Fig. A2) as our threshold.

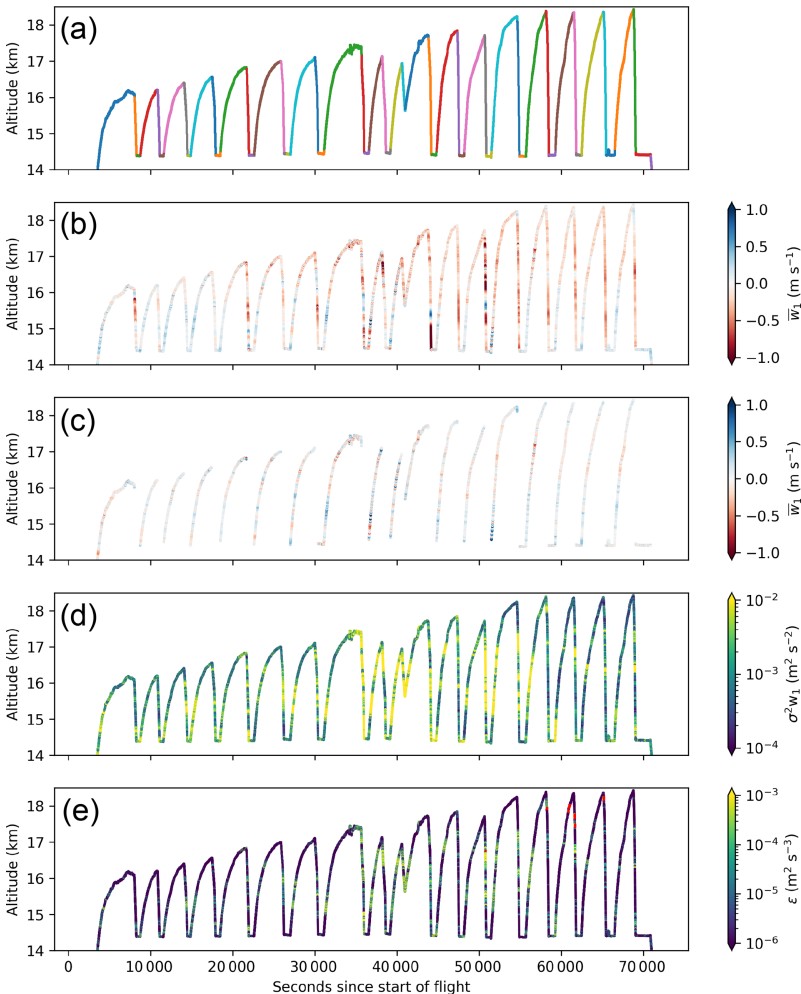

**Figure A1.** Flight tracks from a flight during ATTREX 2014 shown in time–height space, with the color indicating **(a)** different flight legs, **(b)** uncorrected mean 1 s vertical wind ($\overline{w}_1$), **(c)** corrected $\overline{w}_1$, **(d)** high-frequency ($> 1$ Hz) vertical wind variance ($\sigma^2 w_1$), and **(e)** turbulent eddy dissipation rate ($\epsilon$). Outliers in $\epsilon$ are shown in red.

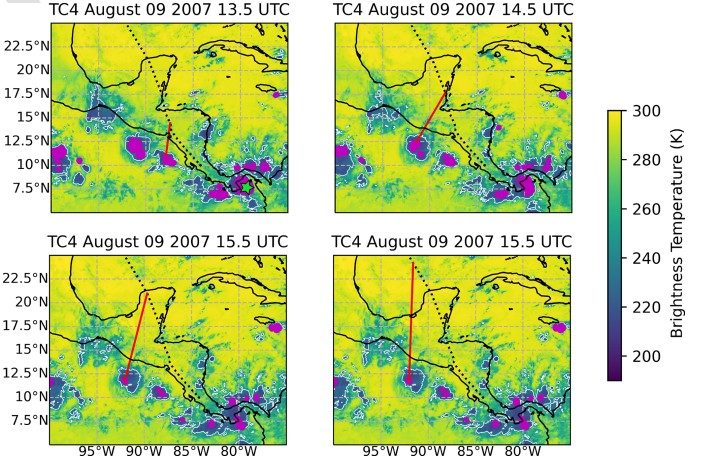

**Figure A2.** Four snapshots of brightness temperature are shown that overlap an example flight from TC4. The dotted black line shows the flight track. In each snapshot, a red line extends between a point along the flight track that is closest in time to the snapshot shown and the nearest deep convective core to that point. Pink dots indicate convective cores with brightness temperatures below 210 K. The green star in the upper left plot is the only point that is below 200 K. The white contours surround areas with brightness temperatures below 235 K.

## Appendix B: Vertical wind variability in all campaigns: additional information

### B1  Data categories

Figure B1 shows histograms of (a) IWC, (b) NI, (c) distance to deep convection, and (d) altitude for all flight campaigns separately. Dotted lines show the boundaries between the categories used in the analyses in Sect. 3.1, 3.2, and 3.3. In general, categories are chosen so that each campaign spans at least two categories, making comparisons between categories more fruitful. NI is only shown for ATTREX 2014 and POSIDON, as the other campaigns do not have enough NI data to support a meaningful analysis.

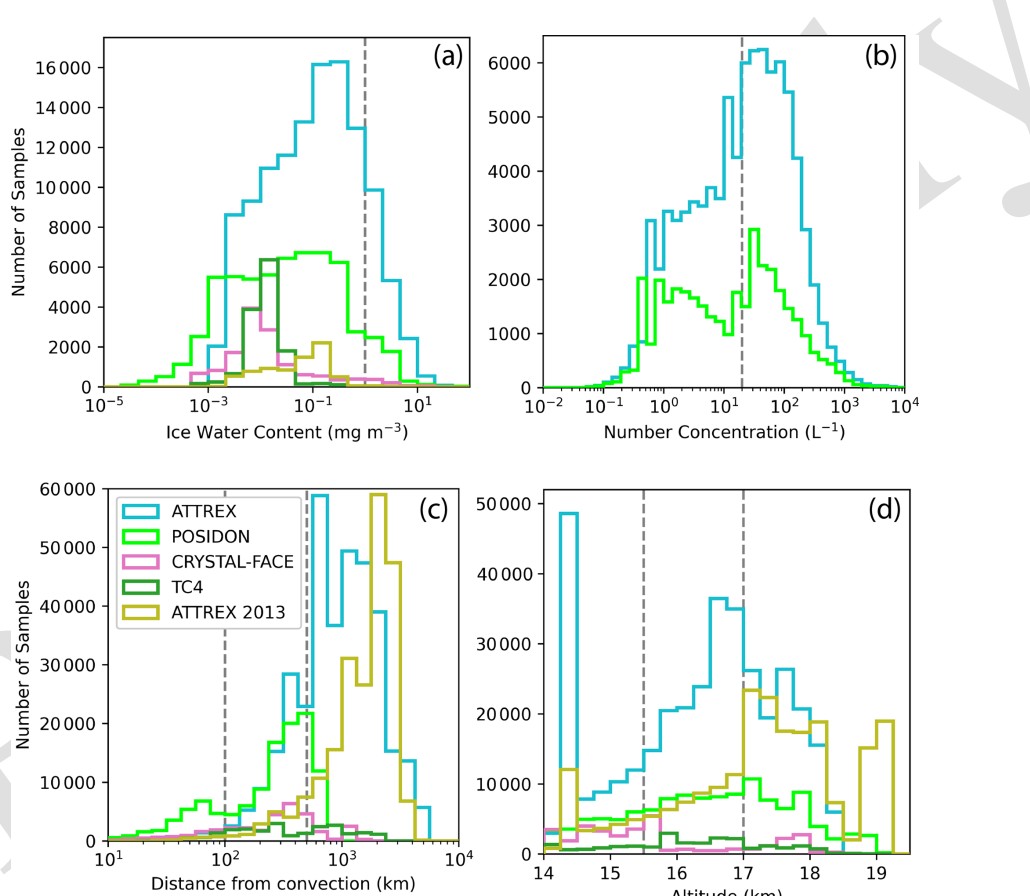

**Figure B1.** Histogram of **(a)** IWC, **(b)** NI, **(c)** distance from deep convection, and **(d)** altitude for all flight campaigns separately. NI is only shown for ATTREX 2014 and POSIDON. Dashed gray lines indicate boundaries used in the study to define categories.

## B2    Clear-sky analysis

Figure B2 shows distributions of $\overline{w}_1$ and $\sigma^2 w_1$ broken up into categories based on IWC and distance from deep convection (like in Figs. 3 and 4) but for clear-sky data only. This analysis shows that the increased vertical wind variability closer to deep convection and lower down in the atmosphere that is shown and discussed in Sect. 3.1 and 3.2 is also seen in the clear-sky data.

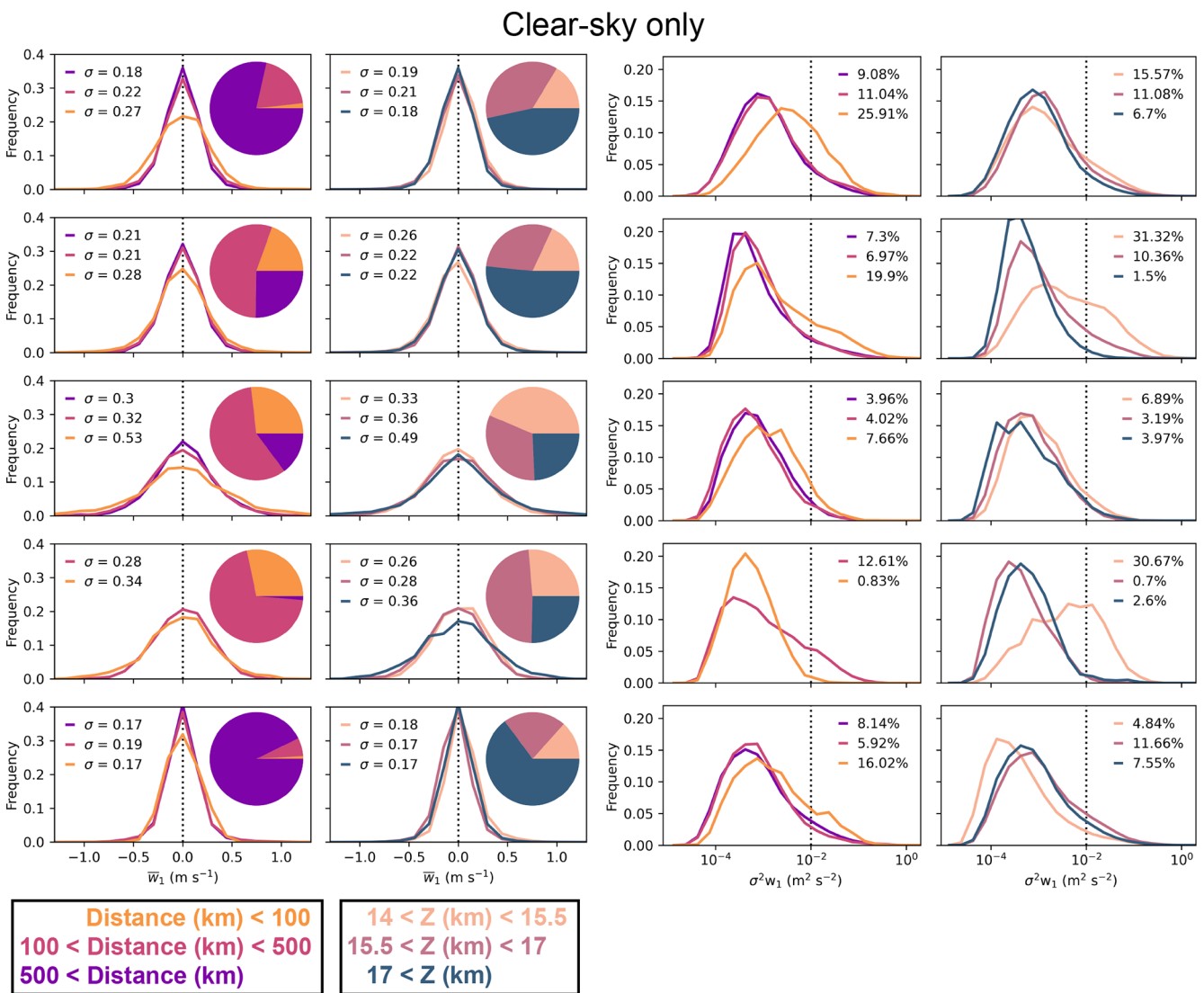

**Figure B2.** Columns 1–2 are the same as columns 3–4 in Fig. 3 but for clear-sky data. Columns 3–4 are the same as columns 3–4 in Fig. 4 but for clear-sky data.

## B3 Turbulent kinetic energy dissipation rate ($\epsilon$)

Figure B3 shows distributions of $\epsilon$ broken up into categories based on IWC, distance from deep convection, and altitude (like in Figs. 3 and 4). This analysis shows that the relationships discussed in Sect. 3.1 between the amount of turbulence and IWC, distance from deep convection, and altitude are robust across different definitions of turbulence, including this much stricter one. Additionally, this figure can be directly compared with Fig. 6 in Podglajen et al. (2017).

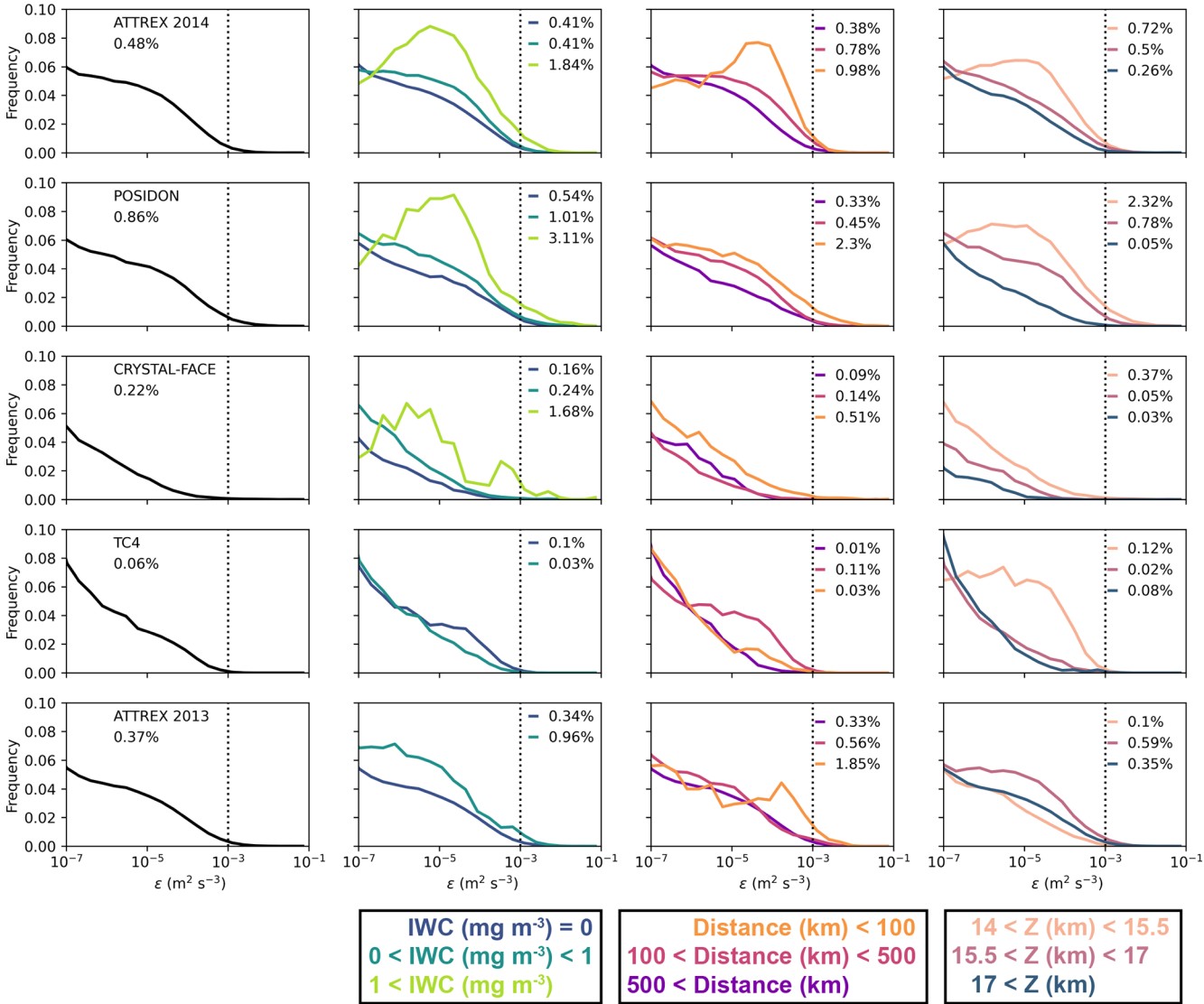

**Figure B3.** Same as Fig. 4 but with distributions of $\epsilon$ instead of $\sigma^2 w_1$ as a proxy for turbulence. A threshold of $10^{-3}$ m$^2$ s$^{-3}$ is used as the turbulence threshold.

## Appendix C: Gravity wave activity and turbulence detection algorithm

We detect turbulence, LW GWA, and SW GWA as follows. For each 25 s/5 km segment within a level leg, we compute the mean vertical wind ($\overline{w}_{25}$), the variance in the high-rate vertical wind ($\sigma^2 w_{25}$), and the mean high-frequency ($> 1\,\mathrm{Hz}$) vertical wind variance ($\overline{\sigma^2 w_1}$; the average of 25 samples of $\sigma^2 w_1$). Thus, for each level leg, we have at least 20 different samples of these variables.

We classify LW GWA based on the difference between the maximum and minimum $\overline{w}_{25}$, which we refer to as $\Delta[\overline{w}_{25}]$, over an entire level leg. If $\Delta[\overline{w}_{25}] > 0.5\,\mathrm{m\,s^{-1}}$, then we classify the level leg as having LW GWA. Otherwise, we classify the level leg as having negligible LW GWA.

In turbulent conditions, the power spectrum of vertical wind in wavenumber $k$ is proportional to $k^{-5/3}$ within the inertial sub-range. Figure C1 shows an example of power spectra for 5 km segments with short-wavelength gravity wave activity (bottom left) and turbulence (bottom right). The spectra are assumed to follow $k^{-5/3}$ behavior in the inertial sub-range, and a proportionality constant (related to the turbulent dissipation rate) between the power spectra and $k^{-5/3}$ is fitted to the parts of the spectra between 20 and 100 m, which is approximately the part of the inertial sub-range that can be resolved with 20 Hz (10 m) vertical wind measurement. The cyan and red lines show the predicted power spectra from those fits. In the turbulent case, the power at scales $> 1$ km is less than what is predicted by $k^{-5/3}$ because those length scales are outside of the inertial sub-range. In the short-wavelength gravity wave case, the power at wavelengths $> 1$ km is greater than what is predicted by $k^{-5/3}$. We interpret that as being due to gravity wave activity at these wavelengths.

The vertical wind variance integrated across wavelengths shorter than $l = 2\pi/k$ is proportional to the integral of the power spectrum across wavenumbers greater than $k$, which is proportional to $k^{-2/3}$ or $l^{2/3}$. Thus, if the ratio of the vertical wind variance across the 5 km (25 s) segment ($\sigma^2 w_{25}$) to the mean vertical wind across 200 m (1 s) sampling windows ($\overline{\sigma^2 w_1}$) exceeds $25^{\frac{2}{3}}$ or 8.5, then we are unlikely to be sampling just turbulence because the variance in vertical wind is increasing more sharply with wavelength than is plausible for turbulence. In the likely event that one or both of these wavelengths is too long to be in the inertial sub-range of the turbulence, the spectral power will decrease more slowly than $k^{-5/3}$, and the 8.5 ratio threshold is still a sufficient condition that the vertical motions are not just due to turbulence. This can be visualized using Fig. C1. The integral of the power spectra up to the dashed blue line is proportional to $\overline{\sigma^2 w_1}$. The integral of the entire power spectrum is proportional to $\sigma^2 w_{25}$. If the power spectra lay along the cyan and red lines, which are the fits to $k^{-5/3}$, then $\sigma^2 w_{25}/\overline{\sigma^2 w_1}$ would be exactly 8.5. In the short-wavelength gravity wave case, the power spectrum is steeper than the cyan line and the ratio is 132. In the turbulence case, the power spectrum is less steep than the red line, and the ratio is 3.

Hence, we classify 5 km segments as SW GWA if $\sigma^2 w_{25} > 0.04\,\mathrm{m^2\,s^{-1}}$ and $\sigma^2 w_{25}/\overline{\sigma^2 w_1} > 9.0$. Together, these conditions define a situation in which there is a large amount of vertical wind variability on length scales smaller than 5 km that cannot be explained by turbulence. We use an empirically chosen threshold of 9.0 instead of 8.5 to be slightly conservative in classifying segments as SW GWA.

For 5 km segments that do not have detectable SW GWA, we check for the presence of turbulence. If $\overline{\sigma^2 w_1} > 0.01\,\mathrm{m^2\,s^{-2}}$, then we classify the segment as turbulent.

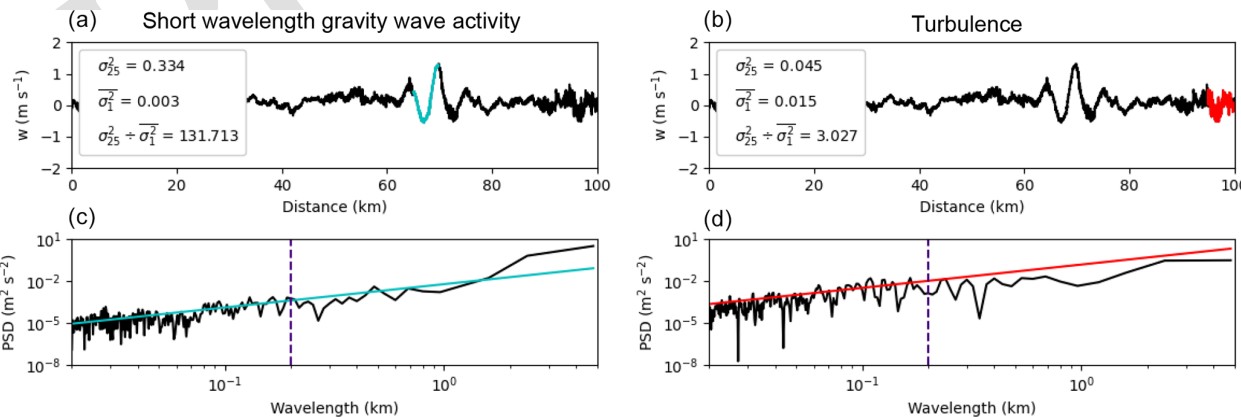

**Figure C1. (a, b)** Vertical wind time series from example 4 in Fig. 5, with specific 5 km sections of interest highlighted in cyan **(a)** and red **(b)**. $\sigma^2 w_{25}$, $\overline{\sigma^2 w_1}$, and $\sigma^2 w_{25}/\overline{\sigma^2 w_1}$ for the highlighted sections are printed on the plots. **(c, d)** Power spectral density for the highlighted 5 km sections (black), with lines fitted to $k^{-5/3}$ between 20 and 100 m (cyan and red).

**Data availability.** High-rate vertical wind measurements for ATTREX 2013–2014 and POSIDON are available on NASA's Earth Science Project Office (ESPO) archive (https://espoarchive.nasa.gov/; ESPO, 2022). Data from CRYSTAL-FACE and TC4 must be requested from T. Paul Bui (thaopaul.v.bui@nasa.gov). Microphysical measurements for all campaigns are available through NASA's ESPO archive and for all campaigns, except ATTREX 2013, at https://doi.org/10.34730/266ca2a41f4946ff97d874bfa458254c (Krämer et al., 2020a). Brightness temperature data are available at https://doi.org/10.5067/P4HZB9N27EKU (Janowiak et al., 2017).

**Supplement.** Video S1 cycles through snapshots of vertical wind for every hour in the DYAMOND-1 simulations, except hours 531–549, for which no vertical wind data are available for ICON. Winds are taken from the model level closest to 14.2 km, and the tropical west Pacific region is shown. The supplement related to this article is available online at: https://doi.org/10.5194/acp-23-1-2023-supplement.

**Author contributions.** RA did all analysis and wrote and edited the paper. CSB provided guidance on the analysis and edited the paper.

**Competing interests.** The contact author has declared that none of the authors has any competing interests.

**Disclaimer.** Publisher's note: Copernicus Publications remains neutral with regard to jurisdictional claims in published maps and institutional affiliations.

**Acknowledgements.** Rachel Atlas acknowledges National Science Foundation (NSF) funding (grant no. OISE-1743753), and Christopher S. Bretherton acknowledges funding from AI2. We are extremely grateful to T. Paul Bui, for providing high-rate MMS data for CRYSTAL-FACE and TC4, and to Rei Ueyama and Jonathan Dean-Day, for their guidance on processing and interpreting the MMS data. We thank Martina Krämer, for guidance on the microphysics guide to cirrus clouds, Jacqueline Nugent and Sami Turbeville, for providing vertical wind data from DYAMOND-1, and Aurélien Podglajen, for helpful commentary on our analysis.

**Financial support.** This research has been supported by the National Science Foundation (grant no. OISE-1743753) and the Allen Institute for Artificial Intelligence (AI2).

**Review statement.** This paper was edited by Stefano Galmarini and reviewed by two anonymous referees.

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

## Remarks from the language copy-editor