# Peer review of "Aircraft observations of gravity wave activity and turbulence in the tropical tropopause layer: prevalence, influence on cirrus and comparison with global storm-resolving models"

_Atmospheric Chemistry and Physics, 2022_

## Author Comment (AC1)

We thank the editor and our two reviewers for their incisive and thorough feedback. We have majorly revised our manuscript in response to the reviewer comments and we believe that it is stronger now. All line numbers here refer to the edited manuscript.

**Reviewer 1**

**General comments**

*The manuscript analyses aircraft observations as obtained from five different measurement campaigns to study the presence of gravity waves and turbulence in the tropical tropopause layer. The study explores relations between the occurrence of gravity waves and turbulence and the distance from deep convective clouds, as diagnosed from satellite images, and the presence of cirrus clouds. The work confirms some existing results, but also highlights some important new ones such as the correlation between gravity waves and turbulence with the microphysical properties of cirrus. The observations are also used to demonstrate that the spectral power of the vertical wind is lower in results from high resolution storm resolving models.*

*The analyses are clearly explained, the paper reads well and the topic is of interest to the readers of ACP. I only have a few minor suggestions. I recommend to accept the manuscript for publication in ACP*

**Suggestions**

- **line 13:** *"Consistent with a previous study". Mention the kind of study (observations/modeling)?*

  We have specified that the previous study is an observational one, using data from two of the aircraft campaigns that we analyze.

- **line 41:** *"vertical motions strongly influence TTL cirrus microphysics, by initiating new instances of homogeneous freezing". Could you give a short physical explanation how/why this could work?*

  We greatly appreciate this suggestion because this mechanism is so important for this work. We have added two paragraphs explaining this to the introduction (lines 49-61).

- **line 99:** *"Flight legs that are less than 100 km long and for which the assumption of small mean wind may be less applicable are removed". What could be reason for the presence of high mean vertical wind velocities, is this possibly due to measurement errors or could there be a physical mechanism involved?*

  We have edited this part of the text (lines 120-121) for clarity. There are physical mechanisms that can cause vertical winds on the order of centimeters per second over 100 km long segments. One example is topographically generated gravity waves, although our aircraft measurements are typically obtained far from topography. However, these large-scale motions are more difficult to measure than small-scale motions of similar magnitudes.

- **line 111:** *It is stated that the constant of proportionality (alpha) in the spectral power relation ($P_w = \alpha \varepsilon^{2/3} k^{-5/3}$) is dependent on the aircraft speed and the sampling frequency. Is there a reference for the reader who is interested in these details?*

  We have added a citation here- thank you for the suggestion.

- **l294:** *wave momentum flux. is it possible to determine momentum fluxes from the observations similar to obtaining the vertical velocity variances? Likewise for the heat flux? Are the*

*turbulent patches also characterized by enhanced variances for the horizontal velocity and temperature?*

There is enhanced variance in the temperature during turbulent periods (as nicely demonstrated in Figure 5 from Podglajen et al. 2017). However, the response time of the temperature probe is too slow for the measurement to be trustworthy on turbulent length scales/timescales so heat fluxes cannot be reliably estimated. The horizontal wind variance is reliable on turbulent scales and was actually used along with the vertical wind variance to identify turbulence in Podglajen et al. 2017. However, this does not mean that momentum fluxes can be reliably estimated. Although we have not tried to compute momentum fluxes for this dataset, we know that in other campaigns, a lack of synchronicity between the u and v measurements made computing momentum fluxes impossible.

- **Fig. C1, line 425:** *it appears that bottom right and bottom left plots have been swapped (bottom left = short wavelength gravity and not turbulence)?*

    Thank you for noticing this. It has been corrected!

**Reviewer 2**

**Summary and general comment**

*In this study wind measurements are used for investigations of gravity waves and turbulence in ice clouds in the tropical tropopause layer. The data from aircraft measurements at different campaigns are separated by several criteria (high/low ice water content, distance to convective activity) in order to determine the impact or relation to ice clouds.*

*Generally, this is a well written manuscript that constitutes a valuable contribution to ACP. However, I have some concerns on the data evaluation of the aircraft measurements and the use of model data. Therefore I recommend major revision for the manuscript, before it can be accepted. In the following I will explain my concerns in detail.*

**Major points**

- **(1) Discrimination between low/high ice water content**

    *In the evaluation of the ice water content a fixed threshold of 1e-3 g/m³ is used. The choice for this constant value is not clearly motivated. Since the water vapor mixing ratio as well as the resulting ice water content (IWC) of ice clouds is highly depending on temperature, it is more than questionable if a fixed threshold is meaningful. The authors mention the investigations by Krämer et al. (2016, 2020) for their interpretation. However, in these publications (e.g. Krämer et al. 2020, fig. 6) the high variation is clearly shown together with a formerly derived parameterization for the median of IWC. It can be seen, that the median value varies over the relevant temperature range 185K<T<200K. I suggest to revise the investigation using a variable threshold (e.g. using the median, formulated in Schiller et al., 2008) or even to investigate the sensitivity of the results due to changes in the threshold. It remains still questionable, if the discrimination is meaningful, since the detection limit of some hygrometers is close to these values (as stated in the text).*

    We performed a sensitivity test to the ice water content threshold for Section 3.1, comparing multiple fixed ice water content thresholds with multiple temperature-dependent ice water content thresholds. In Figure 1 here, the first and third panels show distributions of mean

[Figure]

Figure 1: Sensitivity tests to ice water content threshold. The first and second panels show distributions of mean vertical wind and the third and fourth panels show distributions of vertical wind variance. The lines of the first and third panels correspond to various fixed ice water content thresholds, and the lines on the second and fourth panels correspond to various temperature dependent ice water content thresholds.

vertical wind and vertical wind variance, respectively, for fixed ice water content thresholds which are listed in the legend in the bottom left plot. The brown line matches the ice water content threshold used in the manuscript. The second and forth panels show distributions of mean vertical wind and vertical wind variance, respectively, for ice water content thresholds that vary with temperature. Each threshold is a percentile of ice water content that has been computed for 2 K bins between 184 and 210 K using all campaign data. The percentiles are listed in the bottom plot of the second panel.

The main takeaway from this sensitivity test is that the widening of the mean vertical wind histogram which is seen in ATTREX 2014, POSIDON, and CRYSTAL-FACE, and the shift of the vertical wind variance histogram to the right, seen in all campaigns, for in-cloud data compared to clear-sky data, is evident with all of the IWC thresholds tested. When a very high threshold is chosen, more dramatic shifts can sometimes be seen in ATTREX 2014, POSIDON and CRYSTAL-FACE. However, the thresholds where these more dramatic shifts occur are different for the different campaigns.

The median IWC ranges between 0.012 and 0.31 mg m$^{-3}$ within this temperature range. Therefore, the median IWCs are sometimes below the detection threshold for CRYSTAL-FACE and TC4, which is 0.1 mg m$^{-3}$. As a result, we continue to use our fixed threshold of 1 mg m$^{-3}$, which is well above the detection threshold for all campaigns.

We have mentioned this sensitivity study in lines 190-192 and 221-222.

Our hypothesis, based on the many modelling studies that have preceded our work, is that small-scale vertical motions increase ice crystal number concentrations (NI). We used IWC as a proxy for NI because CRYSTAL-FACE doesn't have reliable NI measurements. In Section 3.3 (Figure 6), we have added NI to our analysis so that our results are more directly applicable to our hypothesis, and less dependent on our analysis of IWC. In Figure 6, we removed the plot comparing IWC with NI because it is less relevant now that we are analyzing NI directly. We replaced it with histograms of distance from deep convection, NI, and IWC where the area of the histogram is filled according to the classified atmospheric conditions of the 5 km segments. These histograms show that the fraction of data that is turbulent increases with increasing IWC and NI, and decreases with increasing distance from deep convection. The fraction of data that is quiescent has the opposite behavior. Thresholds are used again in Figure 5c to simplify the analysis, but Figure 5b shows that the results are not highly sensitive to the thresholds used.

- **(2) Interpretation of low values of ice water content and saturation ratios**

*In section 2.2, an interpretation of values of IWC and ice crystal concentration (NI) together with the saturation ratio (or relative humidity over ice) is given, mostly arguing that ice clouds with small IWC cannot obtain water vapor sufficiently enough. The argumentation is mostly hand-waving and not really convincing. For a more precise argumentation one should use the growth equation for ice mass concentration. The growth rate is proportional to the number concentration (NI) and to the mean radius of the crystals (i.e. $m^{(1/3)}$). Thus, for small ice crystals and/or low number concentrations the rates remain low and the relaxation to equilibrium (i.e. Si=1) is slow. For such estimations, I would suggest to plot the measurements in a different way, i.e. NI as x-axis and ICW as y-axis. Straight lines through*

*the origin would represent different mean masses. A simple back of the envelope calculations from the reported values gave me ice crystal mean masses of order m 4.5e-11kg, leading to a mean equivalent sphere radius of r 24μm. However, the spread is quite large, so plotting different mean masses would help for interpretation.*

*The right plot in figure 2 is not convincing for corroborating the interpretation of slow relaxation to saturation for low IWC values, since (a) the median is only slightly enhanced (110% vs. 100%) and (b) for lower values of IWC the median is lower than saturation.*

*I suggest to carefully reconsider this interpretation also in terms of the investigations of the related vertical velocities. It is well known that a persistent vertical updraft will lead to enhanced steady state values of RHi, so this might also play a role for the interpretation, not just the pure microphysical particle properties together with the thermodynamics. Overall, it is not clear to me if such an interpretation is really relevant for the further investigations of turbulence in ice clouds.*

We thank you very much for your helpful suggestions for improving Figure 2a, and we have made the suggested changes. We agree that Figure 2b is difficult to interpret and does not contribute meaningfully to the narrative of the manuscript and we have removed it.

- **(3) Threshold for turbulence**

  *As the authors stated in their manuscript, the choice of the threshold is somewhat arbitrary (and not explained well in the text) and the threshold is lower than previously defined ones (as e.g. in Podglajen et al., 2017). Since the whole evaluation relies on this threshold, I would suggest to investigate the sensitivity of the results with respect to variations of this threshold. Otherwise, the robustness of this investigation is not really convincing.*

  In Section 3.1, a fixed threshold of 0.01 $m^2s^{-2}$ for vertical wind variability is used to simplify the discussion of how the amount of turbulence depends on distance from convection, height in the atmosphere, and the presence of cloud. The exact increases in the amount of turbulence that we cite do depend on our threshold, but most of the relationships we describe (e.g. turbulence increasing in-cloud) do not. Figures 2 and 3 here are versions of Figure 3 in the manuscript where the legends and the vertical dotted lines correspond to stricter thresholds than what we used in the paper of 0.05 $m^2s^{-2}$ and 0.1 $m^2s^{-2}$, respectively.

  The qualitative results (e.g. whether or not the amount of turbulence increases or decreases when you move further from convection) are sensitive to the thresholds in a few cases where a small percentage of data surpasses the strict thresholds. We list these cases below:

    - 1) For ATTREX 2013, when the most strict threshold of 0.1 $m^2s^{-2}$ is used, there is more turbulence between 100 and 500 km from convection, than there is within 100 km from convection or further than 500 km from convection. However, when the less strict thresholds are used, there is the most turbulence within 100 km of convection.

    - 2) For CRYSTAL-FACE, when a threshold of 0.01 or 0.05 $m^2s^{-2}$ is used, there is more turbulence above 17 km than between 15.5 and 17 km, but when a stricter threshold of 0.1 $m^2s^{-2}$ is used, there is more turbulence between 15.5 km and 17 km than there is above 17 km. However, regardless of the threshold used, the amount of turbulence below 15.5 km is much greater than above 15.5 km.

[Figure]

Figure 2: Version of Figure 3 in the manuscript but with the black dotted lines and legends corresponding to a stricter vertical wind variance threshold of 0.05 m$^2$s$^{-2}$.

[Figure]

Figure 3: Version of Figure 3 in the manuscript but with the black dotted lines and legends corresponding to a stricter vertical wind variance threshold of 0.1 m$^2$s$^{-2}$.

- 3) For TC4, when a threshold of 0.01 m$^2$s$^{-2}$ is used, there is more turbulence between 15.5 and 17 km than above 17 km, but when a stricter threshold of 0.05 or 0.1 m$^2$s$^{-2}$ is used, there is more turbulence above 17 km. However, regardless of the threshold used, the amount of turbulence below 15.5 km is much greater than above 15.5 km. For both TC4 and CRYSTAL-FACE, independent of which threshold is used to define turbulence, there is much more turbulence below 15 km than above 15 km.

We have expanded our explanation of why we chose the turbulence threshold that we did (lines 201-212) and we have noted these sensitivity tests in lines 221-222, 228-230, and 234.

We have also added a figure to the appendix that shows the distribution of the turbulent kinetic eddy dissipation rate ($\varepsilon$), with legends stating the amount of data that has $\varepsilon > 10^{-3}$ m$^2$s$^{-3}$. This serves the dual purpose of further testing the robustness of our results, and making our results easier to compare with Podglajen et al. 2017. The results from this analysis match the results using the stricter threshold of 0.01 m$^2$s$^{-2}$ except in two cases:

- For ATTREX 2014, there is an equal amount of turbulence in clear-sky data and in low-IWC cirrus when a threshold of $10^{-3}$ m$^2$s$^{-3}$ is used on $\varepsilon$. Independent of the threshold used, there is more turbulence in high-IWC cirrus than in clear-sky.

- For ATTREX 2013, when a threshold of $10^{-3}$ m$^2$s$^{-3}$ is used on $\varepsilon$, there is more turbulence within 100 km from convection than between 100 and 500 km from convection (in agreement with thresholds of 0.1 and 0.05 m$^2$s$^{-2}$ on vertical wind variance).

- **(4) Use of model data**

*In section 3.4 4 different models are used for comparison of simulated vertical winds with the measurements. Since the models have very different treatments of convection (parameterisations vs. resolving convective events) and the comparison is really difficult and not convincing, the purpose of this section is not clear to me. Either the authors should explain more in details the models/simulations and what they can learn from this comparison or they should delete this section completely (and adjust the title), since at the moment it does not provide additional insights in the relation between Gw/turbulence and ice clouds. For a better comparison and a meaningful evaluation, ice clouds should also be taken into account. E.g. one could investigate if there is a similar relation between ice clouds close vs. far away from convection in the models.*

None of the models have parameterized convection, as stated in lines 319-320. This is an advantage for simulating clouds compared to traditional GCMs. However, Section 3.3 of our paper shows that small-scale motions, which are either unresolved by the GSRMs (turbulence) or at the grey scale (small-scale gravity wave activity), control cirrus cloud microphysics in the real atmosphere. The purpose of the comparison with GSRMs is to show that the cutting edge resolution of GSRMs is still insufficient for simulating these small-scale motions which are important to cirrus clouds in the real atmosphere. We do not look at the relationships between vertical wind variability and ice water content in the models because FV3 does not output both variables on the same grid. Additionally, none of the models predict ice crystal number concentrations so it would be difficult to relate the physics in the models to the hypothesis investigated in Section 3.3. We have added an explanation to the manuscript of why we do not analyze relationships between simulated cirrus and vertical

winds in the models (lines 355-356). We have also expanded the introduction to give the comparison with the models more context (lines 24-31). We also note, both here and in the manuscript (lines 359-362), that if the GSRMs do a good job of representing TTL cirrus microphysical properties despite lacking the small-scale vertical motions that observations tell us control TTL cirrus microphysics, we would know that the models were getting things right for the wrong reasons (with overproduction of ice within the microphysics schemes being the likely culprit).

**Minor points**

- **(1) Units of ice water content**

  *Since the values of IWC in the upper troposphere are quite small, it is more common to use units like ppm, ppmv or mg per m³, see also Krämer et al. (2020). I would recommend to change the units for improving the readability.*

  We have changed all of the IWC units to mg m$^{-3}$.

- **(2) Turbulence in ice clouds due to buoyancy effects**

  *There are some studies indicating that latent heat release and/or radiative heating inside cirrus clouds might lead to buoyant instabilities (see, e.g., Dobbie and Jonas, 2001; Marsham and Dobbie, 2005; Spichtinger, 2014). It is not clear if such effects also might play a role in the TTL, if the stratification leads to potentially unstable layers. At least, such a possibility should be mentioned.*

  We mentioned this briefly in the original manuscript in the introduction, in the description of Dinh et al. 2010 (lines 65-67). Now, we have also added a paragraph about this to the conclusion (lines 394-400).